# Canonical cortical circuits: A unified sampling machine for static and dynamic inference

## Abstract

The brain lives in an ever-changing world and needs to infer the dynamic evolution of latent states from noisy sensory inputs. Exploring how canonical recurrent neural circuits in the brain realize dynamic inference is a fundamental question in neuroscience. Nearly all existing studies on dynamic inference focus on deterministic algorithms, whereas cortical circuits are intrinsically stochastic, with accumulating evidence suggesting that they employ stochastic Bayesian sampling algorithms. Nevertheless, nearly all circuit sampling studies focused on static inference with fixed posterior over time instead of dynamic inference, leaving a gap between circuit sampling and dynamic inference. To bridge this gap, we study the sampling-based dynamic inference in a canonical recurrent circuit model with excitatory (E) neurons and two types of inhibitory interneurons: parvalbumin (PV) and somatostatin (SOM) neurons. We find that the canonical circuit unifies Langevin and Hamiltonian sampling to infer either static or dynamic latent states with various moving speeds. Remarkably, switching sampling algorithms and adjusting model's internal latent moving speed can be realized by modulating the gain of SOM neurons without changing synaptic weights. Moreover, when the circuit employs Hamiltonian sampling, its sampling trajectories oscillate around the true latent moving state, resembling the decoded spatial trajectories from hippocampal theta sequences. Our work provides overarching connections between the canonical circuit with diverse interneurons and sampling-based dynamic inference, deepening our understanding of the circuit implementation of Bayesian sampling.

## 1 Introduction

The brain is bombarded with a continuous stream of sensory inputs conveying noisy and ambiguous information about the world. Since the world is dynamic, the brain must seamlessly track the dynamic evolution of the world. In statistics, inferring time-varying latent states is often modeled via hidden Markov models (HMMs), a process known as *dynamic inference*. How the canonical recurrent neural circuits in the brain infers dynamic latent states is a fundamental question in neuroscience (Pouget et al., 2013). Previous studies have investigated how neural circuits implement dynamic inference through *deterministic* algorithms (Wu et al., 2003; Rao, 2004; Beck & Pouget, 2007; Deneve et al., 2007; Wilson & Finkel, 2009; Pfister et al., 2009; 2010; Ujfalussy et al., 2015; Kutschireiter et al., 2023), where the circuit is either deterministic or its internal noise is non-essential in their theory.

Accumulating evidence, however, suggests that the neural circuits in the brain employ *stochastic* sampling-based algorithms to perform inference (Hoyer & Hyvärinen, 2003; Buesing et al., 2011; Aitchison & Lengyel, 2016; Haefner et al., 2016; Orbán et al., 2016; Echeveste et al., 2020; Zhang et al., 2023; Terada & Toyoizumi, 2024; Sale & Zhang, 2024). The stochastic sampling closely matches the stochastic nature of neural dynamics, characterized by the large, structured neuronal response variability (Shadlen & Newsome, 1998; Churchland et al., 2011; Orbán et al., 2016; Echeveste et al., 2020; Zhang et al., 2023). Despite this, nearly all existing studies on neural circuit sampling focused on *static inference* with fixed posterior over time (Hoyer & Hyvärinen, 2003; Aitchison & Lengyel, 2016; Haefner et al., 2016; Orbán et al., 2016; Echeveste et al., 2020; Zhang et al., 2023; Sale & Zhang, 2024). There is a gap in our understanding about neural circuit sampling algorithms and dynamic inference. Therefore, we seek to unify neural circuit sampling and dynamic inference by investigating how the canonical circuit can implement sampling-based dynamic inference. Moreover, since the brain performs both static and dynamic inference contingent on the performed

task, we further investigate how the same recurrent neural circuit can flexibly implement and switch between both inference modes without modifying its synaptic weights.

In the brain, the canonical cortical circuit is a fundamental building block of the cerebral cortex, and is consists of excitatory (E) neurons and diverse classes of inhibitory (I) interneurons including parvalbumin (PV), somatostatin (SOM), and other types (Adesnik et al., 2012; Niell, 2015; Fishell & Kepecs, 2020; Niell & Scanziani, 2021; Campagnola et al., 2022). Different interneuron classes have unique intrinsic electrical properties and form specific connectivity patterns (Fig. 1A). The diverse interneurons not only keep the stability of the circuit, but may also modulate the computations in E neurons, e.g., switching the circuit's sampling algorithms (Sale & Zhang, 2024). Building off the previous model, it is reasonable to hypothesize that the diverse interneurons may enable circuits to employ sampling to implement both dynamic and static inference and flexibly switch between them.

To investigate how the same neural circuit can realize both static and dynamic inference, we focus on a canonical recurrent neural circuit model with E neurons and two types of interneurons: PV and SOM neurons. Our circuit model is built upon the model in a recent study (Sale & Zhang, 2024) that is *biologically plausible* (reproducing tuning curves of different types of neurons) and is also *analytically tractable* to identify the circuit algorithm. Briefly, the circuit model is based on continuous attractor networks (CANs), a recurrent circuit model widely-used in neuroscience to explain the continuous stimulus feature processing (Ben-Yishai et al., 1995; Zhang, 1996; Knierim & Zhang, 2012; Khona & Fiete, 2022). And PV neurons provide divisive, unstructured global inhibition to maintain the circuit's stability, while SOM neurons contribute subtractive, structured local inhibition to E neurons.

After theoretical analyses of the nonlinear circuit dynamics, we find the circuit implements a mixture of Langevin and Hamiltonian *sequential* sampling to implement both dynamic and static inference of the latent stimulus. We analytically identify how the internal model of the latent stimulus transition is stored in the circuits (Fig. 2D&G, Fig. 3A). Remarkably, once circuit weights are set (but not fine tuned) as one of optimal configurations, the same circuit with fixed weights can flexibly sample both dynamic and static stimulus posteriors under different latent transition probabilities by only changing the **gain** of SOM neurons (Fig. 3G). Specifically, we find that the SOM's gain is composed of two parts, each serving a specific role: one is a speed dependent gain whose increment enlarges the moving speed of latent stimulus in circuit's internal model, the other is an algorithmic "switching" gain that determines the proportion of blending Langevin and Hamiltonian sampling. Moreover, once the circuit can do dynamic inference, it is automatically *backwards compatible* with simpler static sampling, without adjusting its weights. This non-trivial property arises because the parameter space for dynamic inference is a subset of static inference. When increasing the Hamiltonian sequential sampling component by increasing SOM's gain and E weights, the circuit's sampling trajectory will oscillate around the true latent (moving) stimulus (Fig. 3F), resembling the decoded spatial trajectories from hippocampal theta sequences (Wang et al., 2020), further supporting the biological plausibility of the sequential sampling algorithms employed in the circuit.

**Significance.** This study reveals for the first time how the canonical circuit can flexibly implement sequential sampling for both static and dynamic inference via the gain modulation of SOM neurons, unifying the sampling algorithm, dynamic inference, and its own heterogeneous structure. Considering the canonical circuit is the building block of the cortex, the circuit model with clearly identified algorithm has the potential inspire the building block of the next-generation deep networks.

## 2 BACKGROUND: THE CANONICAL RECURRENT CIRCUIT MODEL

We consider a canonical circuit model consisting of E neurons and two classes of interneurons (PV and SOM) (Fig. 1A), whose dynamics is adopted from a recent circuit modeling study (Sale & Zhang, 2024). The model is *biologically plausible* by reproducing tuning curves of different types of neurons (Fig. A1, B) and is *analytically tractable*, allowing us to directly identify the nonlinear circuit's algorithm. Briefly, each E neuron is tuned for a 1D stimulus $z$ with preferred stimulus $z = \theta$ and the preference of all $N_E$ E neurons, $\{\theta_j\}_{j=1}^{N_E}$, tile the whole stimulus space. E neurons are recurrently connected with a Gaussian kernel in the stimulus space (Eq. 1d). Both PV and SOM interneurons are driven by E neurons, but differ in function: PV neurons deliver global, divisive normalization to E neurons (Eq. 1b), whereas SOM neurons provide local, subtractive inhibition (Eq. 1c). The whole circuit dynamics is (see Sec. B for a detailed explanation and construction rationale).

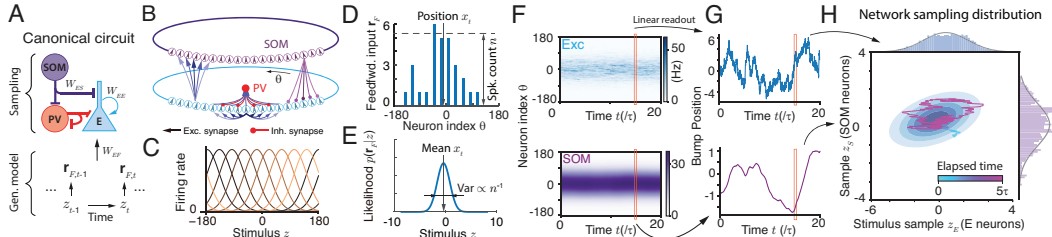

Figure 1: The canonical recurrent circuit model and Bayesian sampling. (A) The canonical neural circuit with E and three types of I neurons receives stochastic feedforward inputs evoked by changing latent stimulus. (B) The recurrent circuit model considered in the present study consists of E and two types of interneurons (PV and SOM). (C) Tuning curves of E neurons. (D-E) A schematic of spiking feedforward input received by E neurons (D) that conveys the whole stimulus likelihood (E). Position and spike count of feedforward input determine the likelihood mean and variance respectively. (F-G) E and SOM population bump responses (F) and instantaneous bump positions (G) regarded as the stimulus sample, $z_E$, and auxiliary variable, $z_S$, respectively. (H) Hamiltonian sampling in the circuit for static stimulus. Panel B-E were adapted from Sale & Zhang (2024) with their permission.

$$\text{E:} \qquad \tau\dot{\mathbf{u}}_E(\theta, t) = -\mathbf{u}_E(\theta, t) + \rho\textstyle\sum_X(\mathbf{W}_{EX} * \mathbf{r}_X)(\theta, t) + \sqrt{\tau\mathsf{F}[\mathbf{u}_E(\theta, t)]_+}\,\xi(\theta, t), \quad \text{(1a)}$$

$$\text{Div. norm.:} \quad \mathbf{r}_E(\theta, t) = [\mathbf{u}_E(\theta, t)]_+^2/(1 + \rho w_{EP}r_P); \quad \text{PV:}\ r_P = \int[\mathbf{u}_E(\theta', t)]_+^2\,d\theta', \quad \text{(1b)}$$

$$\text{SOM:} \qquad \tau\dot{\mathbf{u}}_S(\theta, t) = -\mathbf{u}_S(\theta, t) + \rho(\mathbf{W}_{SE} * \mathbf{r}_E)(\theta, t); \quad \mathbf{r}_S(\theta, t) = g_S \cdot [\mathbf{u}_S(\theta, t)]_+, \quad \text{(1c)}$$

$$\text{Rec. weight:}\ \mathbf{W}_{YX}(\theta - \theta') = w_{YX}\left(\sqrt{2\pi}a_{XY}\right)^{-1}\exp(-(\theta - \theta')^2/2a_{XY}^2), \qquad \text{(1d)}$$

$$\text{Feedfwd.:} \quad \mathbf{r}_F(\theta, t) \sim \text{Poisson}[\lambda_F(\theta|z_t)], \quad \lambda_F(\theta|z_t) = R_F\exp[-(\theta - z_t)^2/2a^2]. \qquad \text{(1e)}$$

where $\mathbf{u}_X$ and $\mathbf{r}_X$ represent the synaptic inputs and firing rates of neurons of type $X$ respectively. In Eq. (1a), the neuronal types $X \in \{E, F, S\}$ representing inputs from E neurons, sensory feedforward inputs (Eq. 1e), and SOM neurons (Eq. 1c) respectively. $[x]_+ = \max(x, 0)$ is the negative rectification. E neurons receive internal Poisson variability with Fano factor $\mathsf{F}$, mimicking stochastic spike generation that can provide appropriate internal variability for circuit sampling (Zhang et al., 2023). In particular, $g_S$ is the "gain" of SOM neurons and can be modulated (see Discussion), which is the key circuit mechanism to flexibly switch between static inference and dynamic inference with various speeds. Details regarding the network can be found in Appendix Sec. B.

To facilitate math analysis, the above dynamics consider infinite number of neurons in theory ($N_E \to \infty$), then the sum of inputs from other neurons $\theta_j$ becomes an integration (convolution) over $\theta$, e.g., $(\mathbf{W} * \mathbf{r})(\theta) = \int \mathbf{W}(\theta - \theta')\mathbf{r}(\theta')d\theta'$, while our simulations take finite number of neurons. $\rho = N/2\pi$ is the neuronal density in the stimulus feature space, a factor in discretizing the integral.

## 2.1 THEORETICAL ANALYSIS OF THE CANONICAL CIRCUIT DYNAMICS

Theoretical approaches to obtain **analytical** solutions of the nonlinear circuit dynamics have been established (Fung et al., 2010; Wu et al., 2016; Zhang & Wu, 2012), including attractor states, full eigenspectrum of the perturbation dynamics, and the projected dynamics onto dominant eigenmodes. These analytical solutions are essential to identify circuit's Bayesian algorithms. Below, we brief the key steps and results of the math analysis, with details in Appendix Sec. D.

**Attractors.** E neurons in canonical circuit dynamics have the following attractor states with a bump profile over the stimulus feature space (Fig. 1F; Sec. D.1),

$$\bar{\mathbf{u}}_E(\theta) = \bar{U}_E\exp[-(\theta - \bar{z}_E)^2/4a^2], \quad \bar{\mathbf{r}}_E(\theta) = \bar{R}_E\exp[-(\theta - \bar{z}_E)^2/2a^2]. \qquad (2)$$

Similar bump attractor states exist for SOM neurons (Eq. D6). In contrast, PV neurons don't have a spatial bump profile since their interactions with E neurons are unstructured (Eq. 1b).

**Dimensionality reduction for stimulus sampling dynamics.** The perturbation analysis reveals that the first two dominant eigenmodes of the circuit dynamics correspond to the change of bump position $z_E$ and the bump height $U_E$ respectively (Sec. D.3, Fung et al. (2010); Wu et al. (2016)). Projecting

the circuit dynamics (Eqs. 1a and 1c) onto the 1st dominant eigenvector (position $z$ change) yields the low-dimensional dynamics of bump positions $z_E$ and $z_S$,

$$\text{E neurons: } \dot{z}_E \approx \tau_E^{-1}[U_{EF}(x_t - z_E) + U_{ES}(z_S - z_E)] + \sigma_E \tau_E^{-1/2}\xi_t, \tag{3a}$$

$$\text{SOM: } \quad \dot{z}_S \approx (\tau_S)^{-1}[U_{SE}(z_E - z_S)] \tag{3b}$$

where $\sigma_E$ is a constant invariant with network activities. The $z_E$ dynamics in Eq. (3a) will be linked to the Bayesian sampling dynamics, i.e., the equilibrium distribution of $z_E$ should match the posterior. Conceptually, this implies the circuit sampling is realized by the fluctuation of the location $z_E$ of the population bump responses, i.e., the fluctuation of population responses along the y-axis in Fig. 1F-G. And the $\bar{z}_E$ in Eq. (2) corresponds to the time-averaged mean of $z_E$. The sampling time constant $\tau_X = \tau U_X$ is proportional to the bump height $U_X$ ($X \in \{E, S\}$), which is solved as (Sec. D.2),

$$(a). \ U_E \approx U_{EE} + U_{EF} + U_{ES}, \quad (b). \ U_S \approx U_{SE}, \quad \text{with } U_{XY} = \rho w_{XY} R_Y / \sqrt{2}. \tag{4}$$

$U_{XY}$ denotes the population input magnitude from neuron type $Y$ to $X$ (see Sec. D.3). Eqs. (3a - 4) are the basis for identifying the circuit sampling algorithms for both static and dynamic inference.

## 3 THE INTERNAL MODEL AND SAMPLING ALGORITHMS IN THE CIRCUITS

The stage from external stimulus feature $z_t$ to the feedforward input $\mathbf{r}_F(t)$ (Eq. 1e) is regarded as the generative process, and then the canonical circuit dynamics is supposed compute the posterior of $z_t$ via its algorithm. We start by proposing an internal generative model in the circuit and assuming it matches with the true world model. Then we will prove our proposition via math analysis.

### 3.1 THE SUBJECTIVE INTERNAL GENERATIVE MODEL IN THE CANONICAL CIRCUIT

**The internal model of latent stimulus transition** stored in the canonical circuit is assumed to have the following form, similar to previous studies (e.g., Deneve et al. (2007); Kutschireiter et al. (2023))

$$p(z_{t+1}|z_t) = \mathcal{N}\big(z_{t+1}|z_t + v\delta t, \Lambda_z^{-1}\delta t\big), \tag{5}$$

where $\delta t$ is the time bin, $v$ is the moving speed of latent stimulus, and $\Lambda_z$ is the precision of the transition. When $v = 0$ and $\Lambda_z \to \infty$, the latent stimulus is static over time, and the stimulus inference degenerates into extensively studied static inference (e.g., Orbán et al. (2016); Echeveste et al. (2020); Masset et al. (2022); Zhang et al. (2023); Sale & Zhang (2024)).

**Stimulus likelihood** $f(z)$. The stochastic feedforward input from the stimulus feature $z$ (Eq. 1e) naturally specifies the stimulus likelihood calculated as a Gaussian (Sec. C.1),

$$f(z_t) \propto p[\mathbf{r}_F(t)|z_t] = \prod_\theta \text{Poisson}[\lambda_F(\theta|z)] \propto \mathcal{N}\big(z_t|x_t, \Lambda_F^{-1}\big), \tag{6}$$

where its mean $x_t = \sum_j \mathbf{r}_F(\theta_j)\theta_j / \sum_j \mathbf{r}_F(\theta_j)$ and precision $\Lambda_F = a^{-2}\sum_j \mathbf{r}_F(\theta_j, t) = \sqrt{2\pi}\rho a^{-1}R_F$ can be read from $\mathbf{r}_F$ via population vector (Georgopoulos et al., 1986; Dayan & Abbott, 2001), and they are geometrically regarded as $\mathbf{r}_F$'s location and the height respectively (Fig. 1D-E). Notably, the Gaussian stimulus likelihood results from the Gaussian profile of feedforward input tuning $\lambda_F(\theta|z_t)$ (Eq. 6) (Ma et al., 2006). And a single snapshot of $\mathbf{r}_F(t)$ parametrically conveys the whole stimulus likelihood $\mathcal{L}(z_t)$ (Ma et al., 2006).

### 3.2 THE PROPOSED SEQUENTIAL SAMPLING FOR DYNAMIC INFERENCE IN THE CIRCUIT

The inference of dynamic latent stimuli has been well established in statistics, and the instantaneous posterior $p(z_t|\mathbf{r}_F(1:t)) \equiv \pi_t(z_t)$ given all feedforward inputs $\mathbf{r}_F(1:t)$ up to time $t$ can be iteratively computed as the recursive Bayesian filtering (RBF, (Bishop, 2006)),

$$\pi_{t+1}(z_{t+1}) \propto f(z_{t+1}) \int p(z_{t+1}|z_t)\pi_t(z_t)dz_t. \tag{7}$$

Although the RBF with Gaussian case permits exact inference via Kalman filter, the circuit implementation of Kalman filter is not straightforward by requiring complex, nonlinear operations (Rao, 2004; Beck & Pouget, 2007) (see more in Discussion). Thus, we consider using **sequential sampling** to approximate the RBF. The flexible representation of sampling can simplify the circuit implementation by eliminating the need for complex, nonlinear operations in deterministic circuits,

which will be shown below. Sequential sampling utilizes stochastic sampling to approximate the integration (marginalization) in Eq. (7), the most challenging computation in Bayesian filtering,

$$
\begin{aligned}
\pi_{t+1}(z_{t+1}) &\propto f(z_{t+1}) \cdot \left[ \tfrac{1}{L} \textstyle\sum_{l=1}^{L} p\big(z_{t+1}|\tilde{z}_t^{(l)}\big) \right], \quad \tilde{z}_t^{(l)} \sim \pi_t(z_t), \\
&\approx f(z_{t+1}) \cdot p\left(z_{t+1}|\tilde{z}_t\right), \quad \text{(only draw one sample, } L=1\text{)}.
\end{aligned}
\tag{8}
$$

Drawing only one sample at each time avoids the need to average over probabilities (Eq. 8), and keeps the posterior $\pi_t(z_t)$ closed as Gaussian, which, otherwise, will be a Gaussian mixture and needs extra approximations in the circuit as in previous studies (Rao, 2004; Beck & Pouget, 2007).

$$
\pi_t(z_t) = \mathcal{N}(z_t|\mu_t, \Omega_t^{-1}), \text{ with } \Omega_t = \Lambda_F + \Lambda_z \delta t^{-1}, \ \mu_t = \Omega_t^{-1}[\Lambda_F x_t + \Lambda_z(\tilde{z}_t \delta t^{-1} + v)].
\tag{9}
$$

It is worth noting that drawing one sample in each time step is a conceptual way of understanding the sampling process in the discrete dynamics. In the continuous case ($\delta t \to 0$), the number of samples that can be drawn in a single "time step" is related to the relative time scales between the latent transition and sampling dynamics. This approximation works well when the latent $z$ changes slowly. In addition, drawing one sample in Eq. (8) doesn't imply the distribution $p\left(z_{t+1}|\tilde{z}_t\right)$ is approximated by a single sample and collapses into a delta function. Rather, it is a parametric Gaussian distribution (Eq. 5) with the mean parameter determined by $\tilde{z}_t$.

### 3.3 NEURALLY PLAUSIBLE SEQUENTIAL SAMPLING DYNAMICS

There are many ways of generating random samples, $\tilde{z}_t \sim \pi_t(z_t)$ (Eq. 8). It was found the canonical circuit can realize both Langevin and Hamiltonian sampling for static inference (Sale & Zhang, 2024), so we consider how to utilize these two sampling dynamics for sequential sampling in dynamic inference and then link them to the circuit's bump position dynamics (Eq. 3a-3b).

**Langevin sequential sampling.** It uses the last sample $\tilde{z}_{t-1}$ to evaluate the gradient of $\pi_t(z)$ and run one step of the Langevin dynamics (Welling & Teh, 2011; Septier & Peters, 2016; Ma et al., 2015),

$$
\tilde{z}_t = \tilde{z}_{t-1} + \left(\tau_L^{-1}\delta t\right)\nabla_z \ln \pi_t(\tilde{z}_{t-1}) + \left(2\tau_L^{-1}\delta t\right)^{1/2}\eta_t,
\tag{10}
$$

with the equilibrium distribution $\pi_t(z)$. In sequential sampling, it is not necessary to draw many samples to reach equilibrium. Instead, one sample is drawn from $\pi_t(z)$ (running Eq. 10 for one step), and then the next sample $\tilde{z}_{t+1}$ is drawn from the next posterior $\pi_{t+1}(z)$. This corresponds to a non-equilibrium Langevin dynamics. The efficiency of the non-equilibrium sequential sampling depends on the time scale of the sampling, $\tau_L$, and the change of posterior $\pi_t(z)$.

**Hamiltonian sequential sampling** defines a Hamiltonian function $H(z,p)$ with a momentum $p$,

$$
H_t(z,p) = -\ln \pi_t(z) + K(p), \quad K(p) = m^{-1}p^2/2.
\tag{11}
$$

$K(p)$ is kinetic energy with $m$ analogous to the mass in physics. We consider the Hamiltonian sampling dynamics with friction $\gamma$ that dampen momentum (Chen et al., 2014; Ma et al., 2015),

$$
\frac{d}{dt}\begin{bmatrix} \tilde{z}_t \\ p_t \end{bmatrix} = -\begin{bmatrix} 0 & -\tau_H \\ \tau_H & \gamma \end{bmatrix}\begin{bmatrix} \nabla_z \ln \pi_t(\tilde{z}_{t-1}) \\ m^{-1}p \end{bmatrix} + \sqrt{2}\begin{bmatrix} 0 & 0 \\ 0 & \gamma^{1/2} \end{bmatrix}\xi_t.
\tag{12}
$$

**Adaptive sampling step size.** Although the step size ($\tau_L$ in Eq. 10) leaves the equilibrium distribution invariant, it is critical in non-equilibrium sequential sampling when only a few samples are drawn from each instantaneous posterior $\pi_t(z)$. The step size determines a trade-off between sampling efficiency and accuracy. For example, a small step size slows down sampling, causing the samples to fail to track fast-moving latent stimuli. Our analysis clearly shows the step size determines a bias and variance trade-off (Sec. E.4). In theory, *Riemann manifold* sampling provides an elegant framework to use $\pi_t(z)$'s curvature, measured by Fisher information $G_t(z)$, to adaptively adjust the step size Amari & Douglas (1998); Girolami & Calderhead (2011); Septier & Peters (2016),

$$
\tau_L \propto G_t(z); \quad m \propto G_t(z); \quad \text{where } G_t(z) = -\mathbb{E}_{\pi_t(z)}\left[\nabla_z^2 \ln \pi_t(z)\right] = \Omega_t.
\tag{13}
$$

Later we will show the circuit can adaptively adjust its sampling step size based on the transition speed and precision of the latent stimulus.

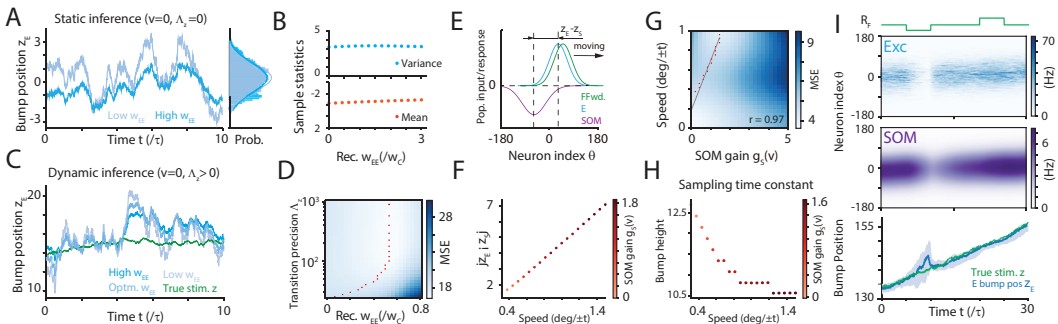

Figure 2: Langevin sampling circuit for both static and dynamic inference. (A-B) The recurrent E weights leave the equilibrium distribution invariant but changing the temporal sampling dynamics. (C) The recurrent weight determines a trade-off between exploitation and exploration, and is critical for non-equilibrium sequential sampling of dynamic inference. (D) The optimal recurrent weight for dynamic inference increases with the latent transition precision. (E-F) The internal speed generation in the circuit comes from the spatial offset of population responses between SOM and E neurons that can be controlled by SOM gain. (G) Higher speed of the latent stimulus requires higher SOM gain in the circuit. (H) The bump height monotonically decreases with the speed as a result of increased gain, yielding a larger sampling step size. (I) The circuit with fixed weights can flexibly sample latent moving stimulus with varying input intensity that controls likelihood precision.

## 4 LANGEVIN SEQ. SAMPLING CIRCUIT FOR DYNAMIC AND STATIC INFERENCE

We validate the circuit implements Langevin sequential sampling with adaptive step size by aligning its bump position dynamics (Eq. 3a) with the Langevin dynamics (Eq. 10). For ease of understanding, we started from a simplified case where the latent stimulus $z_t$ had zero speed but with transition noise ($v = 0$ and $\Lambda_z \neq 0$, Eq. 5). We then extended our solution to the full case with $v \neq 0$.

### 4.1 SAMPLING FOR LATENT STIMULI WITH BROWNIAN MOTION ($v = 0, \Lambda_z \neq 0$)

We compared the circuit sampling dynamics (Eq. 3a) with the Langevin sequential sampling dynamics (Eq. 10) (substituting $\pi_t(z)$ with $v = 0$ in Eq. (9) and step size (Eq. 13) into Eq. (10)) to identify the circuit parameters for Langevin sequential sampling, and the structure of circuit sampling.

|  | Sampling dynamics | Sampling time constant |
|---|---|---|
| E neurons | $\dot{z}_E \approx \tau_E^{-1}[U_{EF}(x_E - z_E)] + \sigma_E \tau_E^{-1/2}\xi_t$ | $\tau_E = \tau U_E = \tau(U_{EE} + U_{EF})$ |
| Langevin | $\dot{z}_t = \tau_L^{-1}[\Lambda_F(x_t - z)] + \left(2\tau_L^{-1}\right)^{1/2}\eta_t$ | $\tau_L \propto \Omega_t = \Lambda_z \delta t^{-1} + \Lambda_F$ |

**Zero-speed latent stimulus requires no SOM inhibition** ($g_S = 0$). When $v = 0$, the Langevin sampling only has a drift term of the likelihood gradient that can be conveyed by the feedforward input in the circuit (comparing the two green terms in the above Table). Hence there is no need for SOM inhibition, realized by a zero SOM gain $g_S = 0$ (Eq. 1c).

**Feedforward E weight $w_{EF}$ is constrained to sample instantaneous posteriors $\pi_t(z)$.** The drift term magnitude $U_{EF} \propto w_{EF} R_F$ (Eq. 4) is a product of the feedforward input strength $R_F$ (proportional to likelihood precision $\Lambda_F$, Eq. 6) and the feedforward weight $w_{EF}$. The $w_{EF}$ needs to adjusted with the $\sigma_E$ to align the sampling variance with the posterior variance.

**Recurrent E weight $w_{EE}$ is constrained by the adaptive sampling time constant $\tau_E$.** $\tau_E$ is critical for non-equilibrium sequential sampling. Comparing the $\tau_E$ and $\tau_L$ in the above Table, since $U_{EF} \propto \Lambda_F$ (Eqs. 6 and 4), the $w_{EE}$ can be set to make $U_{EE} \propto \Lambda_z$ and hence $\tau_E \propto \tau_L$. Fig. 2C shows the bump position trajectory compared with various $w_{EE}$, suggesting $w_{EE}$ is critical in determining the sampling step size. Moreover, each transition precision $\Lambda_z$ has an optimal $w_{EE}$ to minimize the mean square error between the circuit's sample trajectory and the true latent stimulus, and the recurrent weight $w_{EE}$ increases with $\Lambda_z$ (Fig. 2D), confirming our theory (Eq. 14b).

Combined, the circuit parameters for Langevin sequential sampling are (see Sec. E.1.1),

$$(a).\ w_{EF} = \sqrt{\pi}\sigma_E^2/a = (2\sqrt{3})^3\mathsf{F};\quad (b).\ w_{EE} = aw_{EF}(\sqrt{2\pi}\rho\delta t R_E)^{-1}\Lambda_z;\quad (c).\ g_S = 0. \quad (14)$$

**Flexible sequential sampling with various likelihood uncertainties.** Once the circuit paramters are set based on Eq. (14), the circuit can sample $\pi_t(z)$ with various uncertainties without changing synaptic weights (Fig. 2I), all are **automatically** adjusted by the circuit dynamics. Specifically, changing the input intensity $R_F$ changes the likelihood precision $\Lambda_F$ (Eq. 6), which further scales the likelihood gradient magnitude $U_{EF}$ and the bump height $U_E$ determining the sampling step size $\tau_E$.

### 4.2 BACKWARDS COMPATIBILITY FOR STATIC SAMPLING

The static sampling refers to sampling the static posterior $\pi(z) \equiv p(z|\mathbf{r}_F)$ where the latent stimulus $z$ and the given feedforward neural input $\mathbf{r}_F$ are both fixed over time. Although a fixed latent stimulus is realized under static transition parameter ($v = \Lambda_z^{-1} = 0$, Eq. 5), the static posterior $\pi(z)$ is different from the dynamic posterior $\pi_t(z_t)$ obtained under the static parameter. Specifically, the static posterior degenerates into the likelihood, $\pi(z) = f(z)$ (Eq. 6), while the dynamic posterior obtained under static parameter becomes evidence accumulation, i.e., $\pi_t(z_t) = f(z_t)\pi_{t-1}(z_{t-1}) = \prod_t f(z_t)$.

Remarkably, even if the static posterior is different with the dynamic posterior obtained under static parameters, the circuit whose parameters are set for sequential sampling with latent transition noise ($v = 0$ and $\Lambda_z^{-1} \neq 0$, Eq. 14) can automatically sample static posteriors without adjusting any circuit parameters (Fig. 2A). The automatic backwards compatibility arises from the sampling time constant $\tau_E$ (controlled by recurrent E weight $w_{EE}$) is a *free parameter* for static sampling (Eq. 3a) that doesn't affect the equilibrium distribution (Fig. 2B), despite it is critical in sequential sampling (Eq. 14b, Fig. 2D). In static sampling, the convergence time is not an issue and the sampling dynamics can run until equilibrium.

### 4.3 SAMPLING LATENT STIMULI WITH VARIOUS SPEEDS BY MODULATING SOM GAIN

With non-zero latent transition speed $v$, the Langevin sampling has an extra speed $v$-drift term , $\dot{z}_t = \tau_L^{-1}[\Lambda_F(x_t - z) + \Lambda_z v] + (2\tau_L^{-1})^{1/2}\eta_t$. Since $\Lambda_F(x_t - z)$ comes from the feedforward input (Eq. 6), it is straightforward to reason that the inhibitory feedback $U_{ES}(z_S - z_E)$ in the circuit (Eq. 3a) provides the speed-drift term $\Lambda_z v$. We next analyze this in the circuit dynamics.
1) To infer a moving latent stimulus with speed $v$ by the circuit, the E neurons' bump position $z_E$ should move with the same average speed: $\langle \dot{z}_E \rangle = \langle \dot{x}_t \rangle = v$ where $\langle \cdot \rangle$ denoting the average across trials. A similar condition holds for SOM neurons: $\langle \dot{z}_S \rangle = v$ (Eq. 3b).
2) To minimize systematic bias, the average separation between $z_E$ and the input feature $x_t$ should be zero: $\langle x_t - z_E \rangle = 0$, otherwise $\langle z_E(t) \rangle$ is offset from the true latent $z_t$.

All of this implies that the circuit should *internally* generate a moving neural sequence with the same speed as the latent stimulus, rather than being passively driven by the input feature $x_t$. To study the internal speed generation mechanism, we decompose $z_E = \langle z_E \rangle + \delta z_E$, where $\langle z_E \rangle$ is the mean of Eq. (3a) capturing the internally generated movement while the residue $\delta z_E$ captures sampling fluctuation. Substituting the decomposition back to Eq. (3a) yields the dynamics of $\langle z_E \rangle$ and $\delta z_E$,

$$(a).\ \langle \dot{z}_E \rangle = \tau_E^{-1}U_{ES}\langle z_S - z_E \rangle = v;\quad (b).\ \dot{\delta z}_E \approx \tau_E^{-1}U_{EF}(\delta x_t - \delta z_E) + \sigma_E\tau_E^{-1/2}\eta_t, \quad (15)$$

Interestingly, the dynamics of residue $\delta z_E$ correspond to sampling a latent stimulus with zero speed as presented in Sec. 4.1, in that the $\langle \delta x_t \rangle = \langle x_t - \langle x_t \rangle \rangle = 0$.

**Internal speed modulates SOM gain $g_S$.** Combining Eqs. (15a and 4), the circuit's internal speed $v$ is proportional to E and SOM's bump position separation $\langle z_E - z_s \rangle$ (Fig. 2E-F; see Sec. E.1.1)

$$v = \tau^{-1}\langle z_E - z_s \rangle \approx \left(4a^2\tau^{-2}\ln g_s + \text{const.}\right)^{1/2} \quad (16)$$

$\langle z_E - z_s \rangle$, however, is a circuit response rather than an adjustable parameter, so we seek a circuit parameter to modulate the internal speed $v$. We are interested in changing the internal speed by modulating the SOM's gain $g_S$ (Eq. 1c), rather than changing synaptic weights, although both can modulate speed. This is because the neural gain can be quickly modulated at cognitively relevant time scales (tens to hundreds of milliseconds). In contrast, changing synaptic weights requires synaptic plasticity, which is too slow to coordinate with fast cognitive computation.

Considering the weak limit of feedforward input intensity $R_F$, solving Eqs. (3a - 4) obtains a neat relation between $v$ and the SOM gain $g_S$ (Eq. 16, details in Sec. E.1.1). The internal speed increases with the SOM gain $g_S$, a fact confirmed by numerical results (Fig. 2G). In practice, $g_S$ may be modulated by VIP neurons (not included in the present circuit model), conveying the self-motion speed to modulate the internal stimulus speeds (see Discussion).

**Conditions of Langevin sequential sampling for nonzero speed.** Based on the condition for implementing circuit Langevin sampling for a zero speed (Eq. 14), we only need to adjust the SOM gain $g_S$ to make the circuit's internal speed equal to the latent speed (Fig. 2G).

**Faster sampling associated with faster speed.** The circuit has an additional desirable feature, in that it automatically uses a larger sampling step size to sample a faster latent stimulus. This is because the sampling time constant $\tau_E = \tau U_E$ (inversely proportional to the step size) decreases with the SOM gain $g_S$, (Fig. 2H), a feature favored for higher internal speed $v$. By contrast, the speed-dependent sampling step size is not present in the Riemann manifold sampling, in that the Fisher information of the instantaneous posterior $\Omega_t$ does not rely on latent stimulus speed (Eq. 9).

## 5 HAMILTONIAN SEQUENTIAL SAMPLING IN THE CIRCUIT

We further investigate how the circuit implements Hamiltonian sequential sampling (Eq. 12) with SOM gain modulation. Inspired by Langevin sequential sampling in the circuit, we know that inferring a moving latent stimulus requires a deterministic internal speed, realized by the speed-dependent SOM gain $g_S$ (Eq. 16, Fig. 2G). We also know that the residue dynamics $\delta z_E$ correspond to Langevin sampling of a latent stimulus with zero speed (Eq. 15b). Intuitively, upgrading the Langevin to Hamiltonian sampling corresponds to upgrading the Langevin dynamics of $\delta z_E$ (Eq. 15) into Hamiltonian dynamics (Eq. 12). As will be shown, Hamiltonian sequential sampling can be easily realized by increasing **the SOM gain** $g_S$ and **feedforward weight** $w_{EF}$ based on the circuit weights for Langevin sequential sampling in the last section.

We can decompose the bump position dynamics in a similar way, with $z_X = \langle z_X \rangle + \delta z_X$ ($X \in \{E, S\}$) with $\langle z_X \rangle$ capturing deterministic speed. We obtain the residue dynamics of $\delta z_E$ and $\delta z_S$ [1],

$$\tau_E \dot{\delta z_E} = \underbrace{[U_{ES}^H(\delta z_S - \delta z_E) + U_{EF}^H(\delta x_t - \delta z_E)]}_{\text{Momentum } p, \text{ (Hamiltonian)}} + \underbrace{[U_{EF}^L(\delta x_t - \delta z_E) + \sigma_E \sqrt{\tau_E} \xi_t]}_{\text{Langevin}}, \quad (17a)$$

$$\tau_S \dot{\delta z_S} = U_{SE}(\delta z_E - \delta z_S), \quad (17b)$$

where the feedforward input term (Eq. 17a, green) in $\delta z_E$ dynamics is decomposed into two parts reflecting the contribution from the Langevin and Hamiltonian sequential sampling. $U_{EF}^L$ is the feedforward input magnitude in Langevin sequential sampling (the same as in Eq. 15). $U_{ES}^H$ is the extra SOM inhibition to the E neurons, overlaid with the $U_{ES}$ in $\langle z_E \rangle$ dynamics for internal speed generation in Eq. (15). Transforming the above $(\delta z_E, \delta z_S)$ into the $(\delta z_E, p)$ dynamics and organizing it in the standard form of Hamiltonian sampling (Eq. 12) (see Sec. E.2),

$$\begin{bmatrix} \dot{\delta z_E} \\ \dot{p} \end{bmatrix} = - \begin{bmatrix} U_{EF}^L(\tau_E \Lambda_F)^{-1} & -\beta_E \Lambda_F^{-1} \\ \beta_E \Lambda_F^{-1} & \tau_E \beta_p \beta_E \Lambda_F^{-1} \end{bmatrix} \begin{bmatrix} -\nabla_z \ln \pi_t(z_E) \\ (\tau_E \beta_E)^{-1} \Lambda_F \cdot p \end{bmatrix} + \begin{bmatrix} \sigma_E \tau_E^{-1/2} & 0 \\ 0 & \sigma_p \end{bmatrix} \boldsymbol{\xi}_t \quad (18)$$

where $\beta_p$, $\beta_E$ and $\sigma_p$ are functions of the coefficients in Eqs. (17a-17b) (details in Sec. E.2). Eq. (18) shows that the circuit implements mixed Langevin/Hamiltonian sampling rather than pure Hamiltonian sampling, and that the momentum $p$ has a friction term that dampens the momentum.

**Conditions for realizing Hamiltonian sampling.** Realizing sampling in the circuit requires altering the ratio of the drift and diffusion coefficients (Eq. 18). Comparing Eq. (18) to Eqs. (10 and 12), we observe the requirements for realizing Hamiltonian sampling(Sec. E.2),

$$(a). \ w_{EF}^L = (2\sqrt{3})^3 \mathsf{F}; \quad (b). \ (U_S w_{ES}) \cdot g_S^H - R_F \cdot w_{EF}^H = F(w_{EF}^H / w_{EF}^L) U_E, \quad (19)$$

where $F(x)$ is a monotonically increasing function of $x$ and increases with the 1st order of $x$ (see SI Sec. E.2). Condition (a) is already satisfied in the Langevin sampling (Eq. 14a). This suggests that for circuit Hamiltonian sampling, we only need to enlarge the SOM gain $g_S$ and feedforward weight $w_{EF}$, and that the increase of the two is approximately linear (Eq. 19, Fig. A1D).

---

[1] The $U_{ES}(\delta z_S - \delta z_E)$ is negligible in Langevin sampling (Eq. 15) but is essential for Hamiltonian sampling.

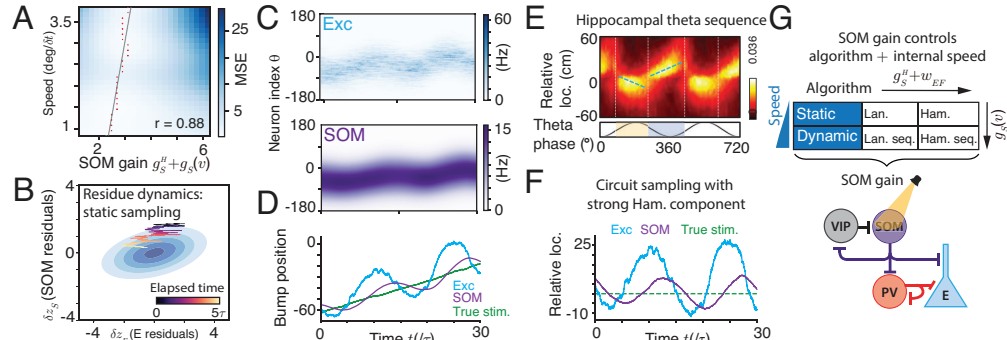

Figure 3: Hamiltonian sequential sampling in the circuit. (A) Additional SOM gain and feedforward weight are needed to upgrade the Langevin sequential sampling into Hamiltonian. (B) The trajectory of the residual dynamics of E and SOM samples resembles sampling a latent stimulus with zero speed. (C-D) Hamiltonian sequential sampling induces oscillations in the E and SOM samples (bump positions) around the true moving latent stimulus. (C): spatiotemporal responses; (D): decoded sampling trajectories. To clearly illustrate the oscillations associated with Hamiltonian sequential sampling, we used a larger SOM gain that deviates from the black line in (A). (E-F) The Hamiltonian sequential sampling is characterized by relative oscillations of samples around the latent moving stimulus (F), and resembles the spatial trajectories decoded during hippocampal theta sequences (E) when a rodent is actively exploring. (E) is adapted from Wang et al. (2020). (G) The SOM gain changes the internal speed of latent stimulus, and switches the sequential sampling algorithms with adjusted with feedforward weights together.

**Hamiltonian sampling trajectory resembles hippocampal theta sequences.** Hamiltonian sequential sampling naturally introduces oscillations in the sampling trajectory $z_E$ (Fig. 3D). For dynamic inference of a moving latent stimulus, oscillations appear as the samples switch between leading and lagging the true latent stimulus (Fig. 3F), while the samples are still centered around the true stimulus. This is comparable to the hippocampus' internal spatial trajectories during theta oscillations when animals are actively exploring the environment (Wang et al., 2020). Furthermore, the hippocampal theta sequences exhibit an asymmetry between the leading and lagging sweeps, where the leading sweep has a larger amplitude (comparing the area above and below zero in Fig. 3E), reproduced in the Hamiltonian sequential sampling in the circuit (Fig 3F). This suggests the biological plausibility of circuit Hamiltonian sequential sampling.

## 6 CONCLUSION AND DISCUSSION

This theoretical study claims canonical recurrent circuits implement sequential sampling to infer either a dynamic or static latent stimulus. It reveals for the first time that the circuit flexibly samples a latent stimulus with different speeds and precisions using either Langevin or Hamiltonian sampling, all of which are modulated by **SOM neurons' gain** ($g_S$, Eq. 1c) containing two parts: one of which increases with the latent stimulus speed (Eq. 16), and the other which acts as a baseline that changes the mixing proportion of Langevin and Hamiltonian sampling (Eq. 19, Fig. 3G). Interestingly, Hamiltonian sequential sampling trajectories in the circuit resemble hippocampal theta sequences (Fig. 3C-F), supporting the biological feasibility of the algorithm.

**What controls SOM's gain?** The SOM receives VIP's inhibition (not included in the model) whose activities are modulated by self-motion signals (Bigelow et al., 2019; Ramamurthy et al., 2023; Guy et al., 2023), possibly by receiving the efference copy from the motor cortex. Therefore, our canonical circuit infers the noisy moving stimulus due to self-motion (see extensions for unknown speed).

**PV gain: an alternative way to adjust internal transition precision.** Our above result considers adjusting the recurrent E weight $w_{EE}$ to modulate the internal transition precision $\Lambda_z$ (Sec. 4.1, Fig. 2D). However, synaptic weight modulation is a slow process that may not be able to capture the rapid change of transition precision. Alternatively, adjusting the internal $\Lambda_z$ can be realized by the fast gain modulation of PV neurons (adding a gain factor $g_P$ in front of $w_{EP}$ in Eq. 1b). The PV's gain determines the firing rate of E neurons ($R_E$ in Eq. 2) that in turn modulate the recurrent E input strength $U_{EE}$ that directly represents the $\Lambda_z$ (Table in Sec. 4.1). Fig. A8 confirms this possibility, where a larger PV gain represents a smaller transition precision.

**Experimental prediction/postdiction**. Sampling a faster latent stimulus needs a larger SOM gain in the circuit (Fig. 3A), causing stronger oscillations of samples (Fig. 3D,F). This reproduces the phenomenon that the theta oscillation power and frequency both increase with movement speed(Whishaw & Vanderwolf, 1973; McFarland et al., 1975; Jeewajee et al., 2008; Hinman et al., 2011; Gupta et al., 2012; Winter et al., 2015; Hinman et al., 2016), and hence our model offers a novel testable circuit mechanism for speed-dependent theta oscillations, whose circuit mechanism remains unclear. For example, a latest circuit model reproduces theta oscillations but is unlikely to reproduce the speed-dependent theta oscillations (Chu et al., 2024).

**Comparison with other inference circuit studies**. **First**, using the same circuit model as Sale & Zhang (2024), we realize a significant step forward in understanding circuit computations. The previous study only considers the static inference (Sale & Zhang, 2024), while we show that the same circuit can flexibly conduct both static and dynamic inference. **Second**, previous studies of dynamics inference neural circuits all considered deterministic circuit algorithms rather than sampling (Wu et al., 2003; Rao, 2004; Beck & Pouget, 2007; Deneve et al., 2007; Wilson & Finkel, 2009; Pfister et al., 2009; 2010; Ujfalussy et al., 2015; Kutschireiter et al., 2023). Depending on the concrete neural representation strategy, deterministic inference circuits require either approximation by interchanging the order of logarithm and sum (Rao, 2004; Beck & Pouget, 2007), or need complicated nonlinear functions to implement the marginalization (Deneve et al., 2007; Wilson & Finkel, 2009; Pfister et al., 2009; 2010; Ujfalussy et al., 2015; Kutschireiter et al., 2023), even if the generative model is linear and exists an exact solution (Kalman filter). The above two challenges are avoided in our sampling circuit: the first by having one effective sample at a time (Eq. 8, $L = 1$), and the second by replacing complex nonlinear functions with stochastic sampling and linear interactions. This result provides a novel benefit of sampling by simplifying the complexity of circuit implementation. **Third**, comparing with other circuit sampling studies (Orbán et al. (2016); Echeveste et al. (2020)) where the sampling is defined in the high-dimensional neural space ($\mathbf{u}$ in Eq. 1a), our circuit sampling is within the low-dimensional stimulus feature subspace (Eq. 3a). Nevertheless, due to the similarity of the circuit models, we believe the sampling defined on different spaces can be unified eventually.

**Distinguish deterministic and sampling circuits in dynamic inference.** The two families of circuit algorithms for dynamics inference can be potentially distinguished in experiments. The Hamiltonian sequential sampling is characterized by the oscillations (Fig. 3D), which are absent in deterministic circuits executing the Kalman filter. To distinguish Langevin sequential sampling from the Kalman filter in neural circuits, we can analyze the variability of circuit estimates (samples). When clamping the posterior $\pi_t(z_t)$ (Eq. 8) and analyzing the samples $z_{t+1}$ in the next time step, the variance of $z_{t+1}$ in Langevin sampling circuit will reflect the variance of $\pi_{t+1}(z_{t+1})$. By contrast, the variance of $z_{t+1}$ in the Kalman filter circuit is either zero (due to deterministic dynamics in the ideal case) or irrelevant with $\pi_{t+1}(z_{t+1})$ (considering irrelevant internal noise in the brain). Practically, claiming the instantaneous posterior can be indirectly realized by analyzing neural data conditioned on the same $z_t$.

**Limitations and extensions of the model**. **First**, although we only present a single circuit model sampling a 1D latent dynamic stimulus, our circuit model and its sequential sampling algorithm can be generalized to sample *multivariate* dynamic latent stimulus (Fig. A4A; Sec. E.3). **Second**, we implicitly assume that the latent speed is provided to the circuit via SOM gain. When the latent speed is unknown, it corresponds to an HMM parameter to be inferred, which can be realized via the forward-backward (FB) algorithms (Bishop, 2006) that naturally require sweepings over the forward and backward directions over time. This FB algorithm is analogous to the oscillations in the Hamiltonian sequential sampling (Fig. 3E-F), suggesting the hippocampal theta sequences might be a candidate circuit mechanism for realizing FB.

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

APPENDIX

## A APPENDIX FIGURES

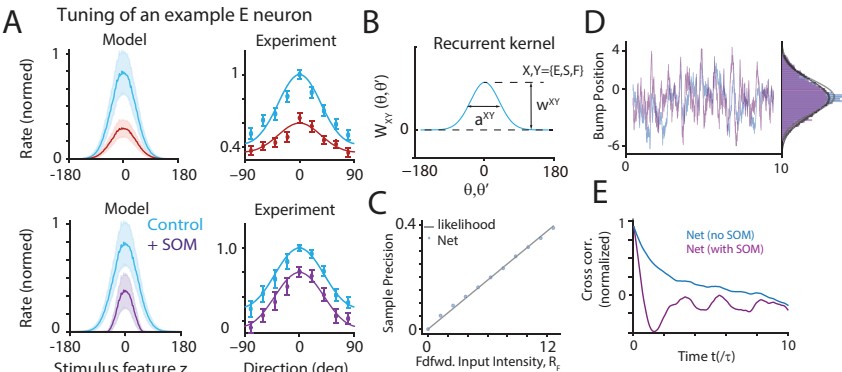

Figure A1: Neuronal tuning curves in the circuit model and static inference. (A) The tuning curve of E neurons in the control state (blue) compared to enhancing PV or SOM neurons. Model generates similar results to experimental data, adapted from Wilson et al. (2012). (B) The recurrent connection kernel in the circuit model. (C) The reduced circuit (without SOM) with fixed parameters flexibly samples different likelihood uncertainties controlled by the feedforward input intensity. (D) The stimulus samples $z_E$ read out from E population responses from the circuit with (purple) and without (blue) SOM neurons can both sample the same likelihood distribution. (E) The circuit with SOM neurons has a faster sampling speed demonstrated by the cross-correlation function of stimulus samples.

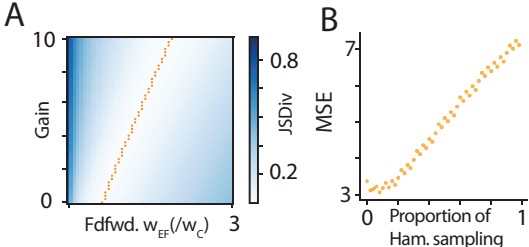

Figure A2: (A) Jensen-Shannon divergence for varying feedforward input weights and SOM gain. There exists a linear manifold where Divergence is minimal. (B) Increasing the proportion of Hamiltonian Monte Carlo (increasing gain and feedforward weight along the manifold of (A) increases the mean square error (MSE).

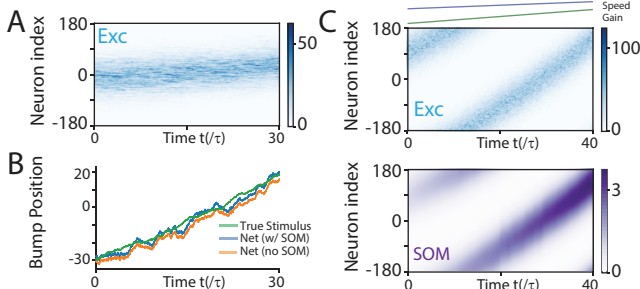

Figure A3: Population activities in response to a moving stimulus with and without SOM neurons. (A) E neuron population responses tracking a moving stimulus. (B) Decoded bump position of E neurons with and without SOM compared with the true latent stimulus. Without SOM neurons, a spatial offset exists between the stimulus sample and true latent stimulus. (C) We select gain-speed pairs from the linear fit of the lowest mean square error (MSE) regime identified in Figs. 2G and 3A. We smoothly increase the latent speed over a $500\tau$ simulation, and use it to generate stochastic input sequences. Meanwhile, we update the SOM gain based on the instantaneous latent speed by using the linear fit of the gain-speed function. We find that the network reliably tracks the moving stimulus with time-varying input speed.

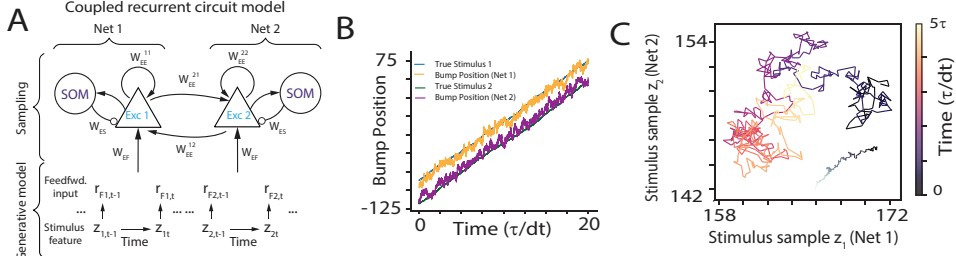

Figure A4: Sampling high-dimensional posteriors. (A) Each latent stimulus, $z_1$ and $z_2$, can be sampled by a recurrent circuit motif that is the same as 1B, coupled together with their coupling storing the prior $p(z_1, z_2)$. (B) Decoded bump position of E neurons from each circuit (net) compared with the true latent stimulus. (C) The 2D sampling trajectories generated by the coupled circuit, and the true latent stimulus (gray line).

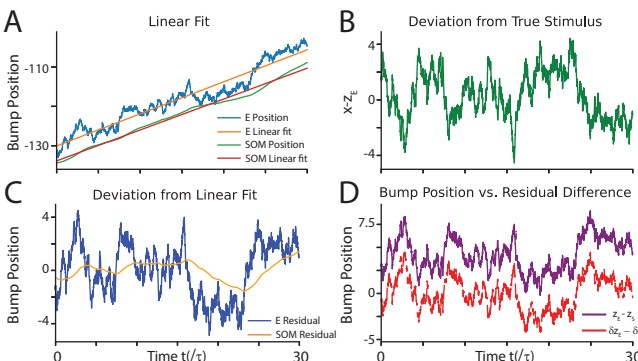

Figure A5: Dynamic inference via Langevin sequential sampling in the reduced circuit with E and PV neurons. (A) Decoded bump position from E and SOM neurons and their respective linear fits in the presence of a moving latent stimulus with non-zero speed. (B) Deviation of the bump position from the true underlying stimulus. (C) Difference between the bump position and fitted linear curve for both SOM and E. Regarded as $\delta z_X$ in main text. (D) Difference of E and SOM bump position (purple) and $\delta z_E - \delta z_S$ (red).

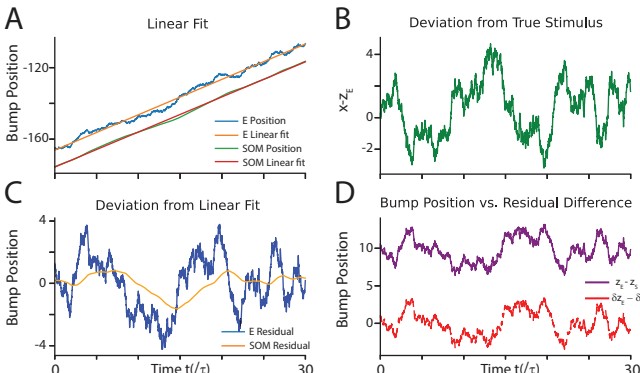

Figure A6: Dynamic inference via Hamiltonian sequential sampling in the augmented circuit with E, PV and SOM neurons. (A) Location of E and SOM population responses and corresponding linear fits. (B) Deviation of read out E position from the true moving latent stimulus. (C). Residuals $\delta z_S$ (orange) and $\delta z_E$ (blue). (D). The separation between E and SOM positions are large while the residual difference $\delta z_E - \delta z_S$ is negligible.

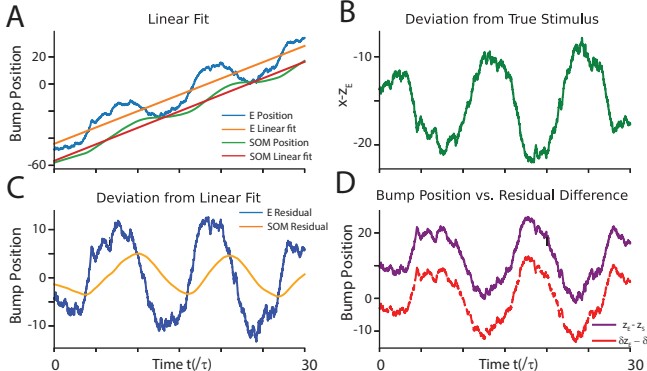

Figure A7: Stimulus and residuals in oscillating network. (A) E and SOM instantaneous population responses with linear regression fit. (B) Difference of E neurons to true latent stimulus. (C) Residuals of E and SOM position to linear fit. (D) Residuals and differences in E and SOM bump position in the oscillating network.

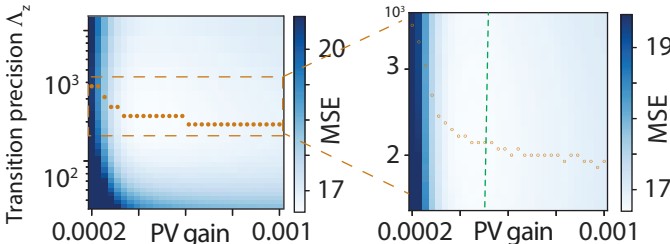

Figure A8: Global inhibition effects bump height. The optimal parvalbumin (PV) gain, $k$, during dynamic inference with $\Lambda_z > 0, v = 0$. A larger $k$ leads to smaller $U_{EE}$ with less transition precision. The fast PV gain modulation also provides a way to change the transition precision in fast time scale. The green dashed line represents the standard PV gain used in the circuit simulations.

## B    THE CANONICAL CIRCUIT MODEL DYNAMICS: DETAILED DESCRIPTION

**E neurons.** Each E neuron is selective for a 1D stimulus feature $z$: the orientation or location. $\theta_j$ denotes the preferred feature of the $j$-th E neuron. The preferred feature of all ,$N_E$, E neurons, $\{\theta_j\}_{j=1}^{N_E}$, uniformly covers the whole space of $z$ (Fig. 1B). Mathematically, in the continuous limit ($\theta_j \to \theta$) corresponding to an infinite number of neurons, the E dynamics is (Zhang et al., 2016; Wu et al., 2008). For convenience, we rewrite the dynamics for E neurons below,

$$\tau \dot{\mathbf{u}}_E(\theta, t) = -\mathbf{u}_E(\theta, t) + \rho \sum_{X \in \{E,F,S\}} (\mathbf{W}_{EX} * \mathbf{r}_X)(\theta, t) + \sqrt{\tau \mathsf{F}[\mathbf{u}_E(\theta, t)]_+} \xi(\theta, t), \quad \text{(B1)}$$

where $\mathbf{u}_E(\theta, t)$ and $\mathbf{r}_E(\theta, t)$ represent the synaptic inputs and firing rates of neurons preferring $z = \theta$, respectively. $X$ denotes neuronal types with $E$, $F$ and $S$ standing for E neurons, sensory feedforward inputs, and the SOM neurons, respectively. $\tau$ is the time constant, $\rho = N/2\pi$ is the neuronal density covering the stimulus feature space, and $[x]_+ = \max(x, 0)$ is negative rectification. E neurons receive internal Poisson variability with Fano factor $\mathsf{F}$, mimicking stochastic spike generation. The symbol $*$ denotes the convolution $\mathbf{W}(\theta) * \mathbf{r}(\theta) = \int \mathbf{W}(\theta - \theta')\mathbf{r}(\theta')d\theta'$. While our math equation (Eq. 1) considers the continuum limit to facilitate math analysis (standard for continuous attractor networks), all simulations are based on a finite number of neurons (e.g., 180 excitatory and 180 SOM neurons). The Gaussian white noise, $\xi(\theta, t)$ is the Dirac delta function $\langle \xi(\theta, t)\xi(\theta', t)\rangle = \delta(\theta - \theta')\delta(t - t)$

**Recurrent weight kernel.** $\mathbf{W}_{YX}(\theta)$ is the recurrent connection kernel from neurons of type $X$ to those of type $Y$, modeled as Gaussian functions (Fig. A1A),

$$\mathbf{W}_{YX}(\theta) = w_{YX}\left(\sqrt{2\pi}a_{XY}\right)^{-1}\exp(-\theta^2/2a_{XY}^2), \quad \text{(B2)}$$

where $w_{YX}$ is the peak amplitude and $a_{XY}$ is the connection width from neuron type $X$ to $Y$. Peak weight from SOM to E is negative for the tuned inhibition ($w_{ES} < 0$). Furthermore, different $W_{XY}(\theta)$ have different connection width, detailed in Sec. F.1.

**PV neurons.** PV neuron firing is driven by E neurons. For simplicity, they are not selective to stimulus, an extreme case of their weak tuning in reality Adesnik et al. (2012); Moore & Wehr (2013); Wilson et al. (2012). PV provides divisive, unstructured global inhibition to E neurons to stabilize the circuit. Their effect is modeled as divisive normalization, a canonical operation (activation function) observed in neural circuits Niell (2015); Carandini & Heeger (2012); Cooke et al. (2020),

$$\mathbf{r}_E(\theta, t) = [\mathbf{u}_E(\theta, t)]_+^2/(1 + \rho w_{EP}r_P), \quad r_P = \int [\mathbf{u}_E(\theta', t)]_+^2 d\theta', \quad \text{(B3)}$$

where $r_p$ (scalar) corresponds to the mean firing rate of the population of PV neurons and $w_{EP}$ (a positive scalar) characterizes the global inhibition strength from PV neurons to E neurons. This divisive normalization function acts as an activation function for the instantaneous synaptic input $\mathbf{u}_E(\theta, t)$ to the firing rate of E neurons $\mathbf{r}_E(\theta, t)$.

**SOM neurons.** In contrast to PV, SOM neurons have tuning and provide subtractive, local structured inhibition to E neurons Wilson et al. (2012). The SOM's dynamics is

$$\tau \dot{\mathbf{u}}_S(\theta, t) = -\mathbf{u}_S(\theta, t) + \rho(\mathbf{W}_{SE} * \mathbf{r}_E)(\theta, t); \quad \mathbf{r}_S(\theta, t) = g_S \cdot [\mathbf{u}_S(\theta, t)]_+. \quad \text{(B4)}$$

where $g_S$ is the "gain" of SOM neurons and can be modulated (see Discussion). As for the E neurons, $\mathbf{u}_S$ and $\mathbf{r}_S$ represent the synaptic input and firing rate for SOM neurons respectively. For simplicity,

we do not consider mutual inhibition between SOM neurons, which negligibly affects our results. Consistent with neuroanatomy, our SOM neurons do not receive direct feedforward input Fishell & Kepecs (2020); Campagnola et al. (2022), which is necessary to realize Hamiltonian sampling as suggested in a recent study Sale & Zhang (2024). Therefore, we only consider SOM neurons being linearly modulated by E neurons, $\boldsymbol{W}_{SE} * \mathbf{r}_E$.

**Stochastic sensory feedforward input.** The feedforward input $\mathbf{r}_F(\theta, t)$ (Eq. 1a) is stochastically evoked from the latent stimulus $z_t$ (Fig. 1D-E), modeled as conditionally independent Poisson spikes with Gaussian tuning given $z_t$, the conventional setting in neural coding studies Ma et al. (2006); Zhang et al. (2023),

$$\mathbf{r}_F(\theta, t) \sim \text{Poisson}[\lambda_F(\theta|z_t)], \quad \lambda_F(\theta|z_t) = R_F \exp[-(\theta - z_t)^2/2a^2], \tag{B5}$$

where $\lambda_F(\theta|z_t)$ is the mean firing rate of neuron $\theta$ given $z_t$. $\mathbf{r}_F$ is approximated as a continuous Gaussian random variable with multiplicative noise.

## C   THE GENERATIVE MODEL AND BAYESIAN SAMPLING

We define a hidden Markov model that consists of the latent stimulus $\{z_t\}$ evoking feedforward inputs $\{\mathbf{r}_F(t)\}$ (Fig. 1A). The latent dynamics evolve as defined by the transition probability,

$$p(z_{t+1}|z_t) = \mathcal{N}(z_{t+1}|z_t + v\delta t, \Lambda_z^{-1}), \tag{C1}$$

where $v$ is the latent transition speed and $\Lambda_z$ characterizes the transition noise. The observed feedforward inputs by the neural circuits are generated as,

$$p(\mathbf{r}_{F,t}|z_t) = \prod_{j=1}^{N_E} \text{Poisson}\left(\mathbf{r}_{F,t}(j)|\lambda_{F,t}(j)\Delta t\right) = \prod_{j=1}^{N_E} \frac{[\lambda_{F,t}(j)\Delta t]^{\mathbf{r}_{F,t}(j)}}{\mathbf{r}_{F,t}(j)!} \exp[-\lambda_{F,t}(j)\Delta t], \tag{C2}$$

### C.1   THE STIMULUS LIKELIHOOD FROM FEEDFORWARD INPUTS

We present the math to determine the stimulus likelihood $p(\mathbf{r}_{Ft}|z_t)$ as previously used in other models Sale & Zhang (2024). The stimulus likelihood function can be derived from the feedforward input by substituting the instantaneous feedforward firing rate, $\lambda_F(\theta|z_t)$, into the Poisson distribution (omitting the time $t$ for clarity). Taking the logarithm of Eq (C2),

$$\begin{aligned} \ln p(\mathbf{r}_F|z) &= \sum_j \left[\mathbf{r}_F(j)\ln(\lambda_F(j)\Delta t) - \ln(\mathbf{r}_F(j)!) - \lambda_F(j)\right], \\ &= \sum_j \mathbf{r}_F(j)\ln(\lambda_F(j)\Delta t) + \text{const.} \end{aligned} \tag{C3}$$

The last line is obtained by assuming the population firing rate, $\sum_j \lambda_F(j)$, is a constant irrelevant with $z$. Substituting the Gaussian profile of feedforward firing rate $\lambda_F(z)$,

$$\begin{aligned} \ln p(\mathbf{r}_F|z) &= -\sum_j \mathbf{r}_F(j)\frac{(\theta_j - z)^2}{2a^2} + \text{const}, \\ &= -\frac{1}{2}\Lambda_F(z - x)^2 + \text{const}, \end{aligned} \tag{C4}$$

where

$$x = \frac{\sum_j \mathbf{r}_F(\theta_j)\theta_j}{\sum_j \mathbf{r}_F(\theta_j)}, \quad \Lambda_F = a^{-2}\sum_j \mathbf{r}_F(\theta_j). \tag{C5}$$

Finally, we approximate the likelihood precision as a function of the peak feedforward firing rate, $R_F$.

$$\begin{aligned} \Lambda_F &\approx a^{-2}\sum_j \lambda_F(\theta_j), \\ &\approx a^{-2}\rho \int \lambda_F(\theta)d\theta \\ &= a^{-2}\rho R_F \int e^{-(\theta-z)^2/2a^2} d\theta \\ &= \sqrt{2\pi}\rho a^{-1}R_F, \end{aligned} \tag{C6}$$

## C.2 The instantaneous stimulus posterior in a hidden Markov model

We are interested in the recursive posterior distribution over the latent state $z_t$ given feedforward inputs up to time $t$ using Bayesian filtering.

$$p(z_{t+1}|\mathbf{r}_{F,1:t+1}) \triangleq \pi_{t+1}(z_{t+1}) = p(\mathbf{r}_{F,t+1}|z_{t+1}) \int p(z_{t+1}|z_t)p(z_t|\mathbf{r}_{F,1:t})dz_t. \tag{C7}$$

Generally, this involves two phases. One is determining the predictive posterior

$$p(z_{t+t}|\mathbf{r}_{F,1:t}) = \int p(z_{t+1}|z_t)\,\pi(z_t)\,dz_t \tag{C8}$$

which estimates how the hidden state changes in the next time step before receiving the observed feedforward inputs. Then the update phase,

$$p(z_{t+1}|\mathbf{r}_{F,1:t+1}) \propto p(\mathbf{r}_{F,t+1}|z_{t+1})\,p(z_{t+1}|\mathbf{r}_{F,1:t}). \tag{C9}$$

which incorporates the latest observation.

## C.3 HMM inference via sequential Sampling

The integral in calculating predictive posterior (Eq. C8) is a generally difficult operation in statistical inference, which imposes a challenge to be implemented in neural circuits.

The present study considers approximately calculating the integral via sampling. Specifically, we replace the integral over $z_t$ with a finite sum of samples $\tilde{z}_t^{(l)} \sim \pi_t(z_t)$, leading to the approximation,

$$\int p(z_{t+1}|z_t)\pi_t(z_t)\,dz_t \propto \frac{1}{L}\sum_{l=1}^{L} p(z_{t+1}|\tilde{z}_t^{(l)}), \quad \tilde{z}_t^{(l)} \sim \pi_t(z_t) \tag{C10}$$

We assume the circuit draws one sample at a time, which simplifies the recursive posterior to,

$$\pi_{t+1}(z_{t+1}) \approx f(z_{t+1}) \cdot p(z_{t+1}|\tilde{z}_t) \equiv \mathcal{N}(z_{t+1}|\mu_{t+1}, \Omega_{t+1}^{-1}) \tag{C11}$$

where

$$\Omega_{t+1} = \Lambda_{F,t} + \Lambda_z$$
$$\mu_{t+1} = \frac{\Lambda_{F,t}\,x_{t+1} + \Lambda_z(\tilde{z}_t + v\Delta t)}{\Lambda_{F,t} + \Lambda_z} = \Omega_{t+1}^{-1}\left[\Lambda_{F,t}\,x_{t+1} + \Lambda_z(\tilde{z}_t + v\Delta t)\right]. \tag{C12}$$

in which we use the sample, $\tilde{z}_t$ drawn from the previous posterior, $\pi_t(z_t)$, to approximate the integral.

**Langevin sampler.** The samples could, for example, be generated by Langevin dynamics, performing a stochastic gradient ascent on the log posterior.

$$\tilde{z}_{t+1} = \tilde{z}_t + \frac{\varepsilon^2}{2}\nabla_z \ln \pi_{t+1}(\tilde{z}_t) + \varepsilon\eta_t \tag{C13}$$

where $\eta_t \sim \mathcal{N}(0, I)$ and the gradient is given by:

$$\nabla_z \ln \pi_{t+1}(\tilde{z}_t) = \Lambda_{F,t}(x_{t+1} - \tilde{z}_t) + \Lambda_z v \tag{C14}$$

In the absence of a changing latent stimulus (i.e., $v = \Lambda_z^{-1} = 0$), the static posterior degenerates to the likelihood,

$$\tilde{z}_{t+1} = \tilde{z}_t + \frac{\varepsilon^2}{2}\Lambda_{F,t}(x_{t+1} - \tilde{z}_t) + \varepsilon\eta_t. \tag{C15}$$

where

$$\frac{d\tilde{z}_{t+1}}{dt} = \lim_{\delta t \to 0} \frac{\tilde{z}_{t+1} - \tilde{z}_t}{\delta t} = \frac{\varepsilon^2}{2\delta t}\Lambda_{F,t}(x_{t+1} - \tilde{z}_t) + \frac{\varepsilon}{\sqrt{\delta t}}\eta_t \tag{C16}$$

Later we will show the Langevin sequential sampling can be implemented by the circuit dynamics for static and dynamic inference.

## D  THEORETICAL ANALYSIS OF THE NONLINEAR CIRCUIT DYNAMICS

### D.1  NETWORK GAUSSIAN ATTRACTOR ANSATZ

We analyze the circuit's attractor state. To ease of reading, we copy the circuit dynamics in below (Eqs 1a and 1c),

$$\tau\frac{\partial \mathbf{u}_E(\theta,t)}{\partial t} = -\mathbf{u}_E(\theta,t) + \rho \sum_{X=E,F,S} (\mathbf{W}_{EX} * \mathbf{r}_X)(\theta,t) + \sqrt{\tau \mathsf{F}_E[\mathbf{u}_E(\theta,t)]_+}\xi(\theta,t),$$

$$\tau\frac{\partial \mathbf{u}_S(\theta,t)}{\partial t} = -\mathbf{u}_S(\theta,t) + \rho \mathbf{W}_{SE} * \mathbf{r}_E(\theta,t); \quad \mathbf{r}_S(\theta,t) = g_S \cdot [\mathbf{u}_S(\theta,t)]_+,$$

(D1)

Taking the equilibrium mean, we have

$$\langle \mathbf{u}_E(\theta) \rangle = \rho \sum_{X=E,F,S} (\mathbf{W}_{EX} \cdot \langle \mathbf{r}_X \rangle)(\theta),$$

$$\langle \mathbf{u}_S(\theta) \rangle = \rho \mathbf{W}_{SE} \cdot \langle \mathbf{r}_E \rangle(\theta),$$

(D2)

We propose the following Gaussian ansatz satisfying the equilibrium state of the circuit dynamics, consistent with previous workSale & Zhang (2024); Zhang et al. (2020),

$$\langle \mathbf{u}_E(\theta) \rangle = U_E \exp\left[-\frac{(\theta - z_E)^2}{4a_E^2}\right].$$

(D3)

We obtain the ansatz of firing rate for E neurons by substituting $\langle U_E(\theta) \rangle$ into divisive normalization,

$$\langle \mathbf{r}_E(\theta) \rangle = \frac{[U_E^2(\theta,t)]^2}{1 + \rho w_{EP} \int [U_E(\theta,t)]^2 d\theta} = \underbrace{\frac{U_E^2}{1 + \rho w_{EP} U_E^2 \sqrt{2\pi}a_E}}_{R_E} \exp\left[-\frac{(\theta - z_E)^2}{2a_E^2}\right].$$

(D4)

Then we can substitute the Gaussian ansatz into the stationary state for the circuit dynamics for each input.

$$\langle I_{XY}(\theta) \rangle = \rho \mathbf{W}_{XY} * \langle \mathbf{r}_Y(\theta) \rangle$$

$$= \rho \int \mathbf{w}_{XY}(\theta' - \theta) \langle \mathbf{r}_Y(\theta) \rangle d\theta'$$

$$= \rho w_{XY} R_Y \frac{a_Y}{\sqrt{a_{XY}^2 + a_Y^2}} \exp\left[-\frac{(\theta - z_Y)^2}{2(a_{XY}^2 + a_Y^2)}\right].$$

(D5)

After substitution, the E and SOM dynamics are then,

$$U_E \exp\left[-\frac{(\theta - z_E)^2}{4a_E^2}\right] = \frac{\rho}{\sqrt{2}}\left[w_{EE}R_E e^{-(\theta - z_E)^2/4a_E^2} + w_{EF}R_F e^{-(\theta - \mu_z)^2/4a_E^2}\right]$$

$$+ \rho w_{ES}R_S \frac{a_S}{\sqrt{2}a_E}\exp\left[-\frac{(\theta - z_S)^2}{4a_E^2}\right].$$

(D6)

$$U_S \exp\left[-\frac{(\theta - z_S)^2}{4a_S^2}\right] = \rho w_{SE}R_E \frac{a_E}{\sqrt{a_{SE}^2 + a_E^2}}\exp\left[-\frac{(\theta - z_E)^2}{2(a_{SE}^2 + a_E^2)}\right],$$

from which we can get the following constraints on the connection width,

$$a_S^2 = a_{SE}^2 + a_E^2$$

$$a_E^2 = a_{ES}^2 + a_{SE}^2.$$

(D7)

Since we have a summation of functions, we can make the approximation that when the positions are sufficiently close together, $z_E = z_S = \mu_z$, then the sum will also be Gaussian. Therefore, the ansatz is adequate.

## D.2 Critical recurrent weight

In our simulations, we scale the peak connect weight in the connection kernel by the smallest recurrent weight where the E network can self-sustain persistent activity without any external input. To find this critical weight, $w_c$, we start with Eq. (D6) in equilibrium,

$$
\begin{aligned}
U_E &= \frac{\rho}{\sqrt{2}}\left(w_{EE}R_E + \frac{a_S}{a_E}w_{ES}R_S\right), \\
U_S &= \frac{\rho}{\sqrt{2}}\frac{a_E}{a_S}w_{SE}R_E.
\end{aligned}
\tag{D8}
$$

Combining $R_S = g_S U_S$ from Eq. (1c) in the main text with Eq. (D8) we obtain,

$$
U_E = \frac{\rho}{\sqrt{2}}R_E\left(w_{EE} + \frac{\rho}{\sqrt{2}}w_{ES}g_S w_{SE}\right)
\tag{D9}
$$

Substituting Eq. (D9) into Eq. (D4)

$$
U_E = \frac{\rho U_E^2}{\sqrt{2} + 2\sqrt{\pi}k\rho a_E U_E^2}\left(w_{EE} + \frac{\rho}{\sqrt{2}}w_{ES}w_{SE}g_S\right),
\tag{D10}
$$

we can find $U_E$,

$$
2\sqrt{\pi}k\rho a_E U_E^2 - \rho U_E \underbrace{\left(w_{EE} + \frac{\rho}{\sqrt{2}}w_{ES}w_{SE}g_S\right)}_{w_c} + \sqrt{2} = 0.
\tag{D11}
$$

with solution

$$
U_E = \frac{\rho w_c \pm \sqrt{\rho^2 w_c^2 - 8\sqrt{2\pi}k\rho a_E}}{4\sqrt{\pi}\rho a_E}
\tag{D12}
$$

For persistent activity, the inside of the square root should be positive. Therefore, we can solve for the critical recurrent weight, $w_c$,

$$
w_c^2 > \frac{8\sqrt{2\pi}k a_E}{\rho}
$$

## D.3 Projection of circuit dynamics on dominant eigenfunctions

We next substitute the Gaussian ansatz into the E and SOM neurons' dynamics.

For the E neurons' dynamics,

$$
\begin{aligned}
\text{LHS} &= \tau U_E \frac{d}{dt}\exp\left[-\frac{(\theta - z_E)^2}{4a_E^2}\right] = \tau\left[U_E\frac{\theta - z_E}{2a_E^2}\frac{dz_E}{dt} + \frac{dU_E}{dt}\right]\exp\left[-\frac{(\theta - z_E)^2}{4a_E^2}\right], \\
\text{RHS} &= \left(-U_E + \frac{\rho w_{EE}R_E}{\sqrt{2}}\right)\exp\left[-\frac{(\theta - z_E)^2}{4a_E^2}\right] + \frac{\rho w_{ES}R_S a_S}{\sqrt{2}a_E}\exp\left[-\frac{(\theta - z_S)^2}{4a_E^2}\right] \\
&\quad + \frac{\rho w_{EF}R_{EF}}{\sqrt{2}}\exp\left[-\frac{(\theta - \mu_z)^2}{4a_E^2}\right] + \sqrt{\tau\mathsf{F}}U_E\exp\left[-\frac{(\theta - z_E)^2}{8a_E^2}\right]\xi(\theta, t)
\end{aligned}
\tag{D13}
$$

Then again, for the SOM neurons,

$$
\begin{aligned}
\text{LHS} &= \tau U_S \frac{d}{dt}\exp\left[-\frac{(\theta - z_S)^2}{4a_S^2}\right] = \tau\left[U_S\frac{\theta - z_S}{2a_S^2}\frac{dz_S}{dt} + \frac{dU_S}{dt}\right]\exp\left[-\frac{(\theta - z_S)^2}{4a_S^2}\right], \\
\text{RHS} &= -U_S\exp\left[-\frac{(\theta - z_S)^2}{4a_S^2}\right] + \frac{\rho w_{SE}R_E a_E}{\sqrt{2}a_S}\exp\left[-\frac{(\theta - z_E)^2}{4a_S^2}\right].
\end{aligned}
\tag{D14}
$$

Previous work has identified two dominant motion modes of recurrent attractor networks Wu et al. (2008); Fung et al. (2010). These two dominant modes correspond to the eigenfunctions of the bump

height and bump position,

$$\text{Position:} \quad \phi_0(\theta|z_X) \propto \exp\left[-\frac{(\theta - z_X)^2}{4a_X^2}\right] \tag{D15}$$

$$\text{Height:} \quad \phi_1(\theta|z_X) \propto (\theta - z_X)\exp\left[-\frac{(\theta - z_X)^2}{4a_X^2}\right]. \tag{D16}$$

where $\phi_0$ and $\phi_1$ are the eigenfunctions of the bump height and position, respectively.

**Projections of the E dynamics**
Then projecting Eq. (D13) onto the bump height eigenfunction $\phi_0(\theta|z_E)$

$$\tau\frac{dU_E}{dt} = -U_E + \frac{\rho}{\sqrt{2}}w_{EE}R_E + \frac{\rho w_{ES}R_S a_S}{\sqrt{2}a_E}\exp\left[-\frac{(z_S - z_E)^2}{8a_E^2}\right]$$
$$+ \frac{\rho}{\sqrt{2}}w_{EF}R_F\exp\left[-\frac{(\mu_z - z_E)^2}{8a_E^2}\right] + \sqrt{\frac{\mathsf{F}}{a_E\sqrt{3\pi}}}\sqrt{\tau U_E}\eta_t \tag{D17}$$

and the position eigenfunction $\phi_1(\theta|z_E)$,

$$\tau U_E\frac{dz_E}{dt} = \frac{\rho w_{ES}R_S a_S}{\sqrt{2}a_E}(z_S - z_E)\exp\left[-\frac{(z_S - z_E)^2}{8a_E^2}\right],$$
$$+ \frac{\rho}{\sqrt{2}}w_{EF}R_F(\mu_z - z_E)\exp\left[-\frac{(\mu_z - z_E)^2}{8a_E^2}\right] + \sqrt{\frac{8a_E\mathsf{F}}{3\sqrt{3\pi}}}\sqrt{\tau U_E}\xi_t \tag{D18}$$

**Projections of the SOM dynamics**
We similarly project the SOM dynamics onto the bump height and position eigenfunction, $\phi_0(\theta|z_S)$ and $\phi_1(\theta|z_S)$, respectively.

$$\text{Position:} \quad \tau\frac{dU_S}{dt} = -U_S + \frac{\rho w_{SE}R_E a_E}{\sqrt{2}a_S}\exp\left[-\frac{(z_E - z_S)^2}{8a_S^2}\right] \tag{D19}$$

$$\text{Height:} \quad \tau U_S\frac{dz_S}{dt} = \frac{\rho w_{SE}R_E a_E}{\sqrt{2}a_S}(z_E - z_S)\exp\left[-\frac{(z_E - z_S)^2}{8a_S^2}\right] \tag{D20}$$

We define the following variables to simplify notations,

$$\tau_X = \tau U_X, \quad U_{XY} = \frac{\rho a_Y}{\sqrt{2}a_X}w_{XY}R_Y, \quad \sigma_E = \sqrt{\frac{8\mathsf{F}a_E}{3\sqrt{3\pi}}}, \quad \sigma_U = \sqrt{\frac{\mathsf{F}}{a_E\sqrt{3\pi}}} \tag{D21}$$

Then, the E and SOM dynamics simplifies to,

$$\tau\dot{U}_E = -U_E + U_{EE} + U_{ES}e^{-(z_S - z_E)^2/8a^2} + U_{EF}e^{-(x - z_E)^2/8a^2} + \sigma_U\tau_E^{1/2}\eta_t \tag{D22a}$$
$$\tau\dot{U}_S = -U_S + U_{SE}e^{-(z_S - z_E)^2/8a^2} \tag{D22b}$$
$$\tau_E\dot{z}_E = U_{ES}e^{-(z_S - z_E)^2/8a^2}(z_S - z_E) + U_{EF}e^{-(x - z_E)^2/8a^2}(x - z_E) + \sigma_E\tau_E^{1/2}\xi_t \tag{D22c}$$
$$\tau_S\dot{z}_S = U_{SE}e^{-(z_S - z_E)^2/8a^2}(z_E - z_S) \tag{D22d}$$

In equilibrium, we assume the difference between bump positions is small enough compared to the connection width $a$, i.e., $|z_E - z_S|$ and $|\mu_z - z_E| << 4a$. Furthermore, we assume the bump height $U_E$ and $U_S$ are large enough, which makes the bump position dynamics (time constant) much slower than the height dynamics, and then we consider the stationary state of the height $U_E$ and $U_S$. In this case, we can simplify the dynamics for the bump height and position,

$$U_E = U_{EE} + U_{ES} + U_{EF} + \sigma_U\tau_E^{1/2}\eta_t \tag{D23a}$$
$$U_S = U_{SE}, \tag{D23b}$$
$$\tau_E\dot{z}_E = U_{ES}(z_S - z_E) + U_{EF}(x - z_E) + \sigma_E\tau_E^{1/2}\xi_t, \tag{D23c}$$
$$\tau_S\dot{z}_S = U_{SE}(z_E - z_S). \tag{D23d}$$

# E  STATIC AND DYNAMIC BAYESIAN SAMPLING IN RECURRENT CIRCUIT DYNAMICS

## E.1  LANGEVIN SEQUENTIAL SAMPLING

### E.1.1  STATIC LATENT STIMULUS (ZERO SPEED, NO NOISY TRANSITION)

To map the network dynamics to Bayesian inference, we start with the static inference case where the network utilizes Langevin sampling, previously described Sale & Zhang (2024). Comparing Eq. (C16) to the circuit dynamics derived in Sec. D.3 without SOM,

$$\dot{z}_E = \tau_E^{-1} U_{EF}(\mu - z_E) + \sigma_E \tau_E^{-1/2} \eta_t. \tag{E1}$$

We can compare $\varepsilon$ to $\tau_E$ to derive the condition in which our network realizes sequential sampling. From Eq. (D21), $\tau_E = \tau U_E$, and $U_E$ has been previously derived as (without SOM),

$$U_E = \frac{\rho}{\sqrt{2}}\big(w_{EE} R_E + w_{EF} R_F\big), \tag{E2}$$

Previous work has mapped the feedforward input to the likelihood distribution in Langevin sampling. In accordance with those derivations, we consider the recurrent weight $w_{EE}$ to be proportional to the latent transition precision, $\Lambda_z$. Then,

$$\varepsilon^2 = U_E \delta t = \frac{\rho}{\sqrt{2}}\bigg[ w_{EE} R_E + w_{EF} R_F \bigg]$$

$$U_E = \underbrace{\frac{\rho w_{EE} R_E}{\sqrt{2}}}_{\propto \Lambda_z} + \underbrace{\frac{\rho w_{EF} R_F}{\sqrt{2}}}_{\propto \Lambda_F}. \tag{E3}$$

Therefore, we consider

$$(a).\ w_{EF} = \sqrt{\pi}\sigma_E^2/a = (2\sqrt{3})^3 \mathsf{F}; \quad (b).\ w_{EE} = a w_{EF}(\sqrt{2\pi}\rho \delta t R_E)^{-1}\Lambda_z; \quad (c).\ g_S = 0. \tag{E4}$$

Dynamic Langevin sampling (non-zero speed and noisy latent transition) When the latent variable changes with constant velocity and transition noise, the transition probability is,

$$p(z_{t+1}|z_t) = \mathcal{N}(z_{t+1}|z_t + v\delta t, \Lambda_z^{-1}). \tag{E5}$$

Starting from the projected dynamics in Eq. (D22), if the circuit can infer the moving latent stimulus accurately, the average speed of the circuit's sample should match the speed of input $x$,

$$(a).\ \langle \dot{z}_E \rangle = \langle \dot{z}_S \rangle = \langle \dot{x} \rangle = v, \quad (b).\ \langle x_t - \langle z_E \rangle \rangle \approx 0. \tag{E6}$$

And the 2nd equality in the above equation is obtained that the average difference between the input $x_t$ and circuit sample $z_E$ should be close to zero, otherwise there will be a systematic bias.

**The internal speed in the circuit**

If we average the bump position dynamics over trials (Eqs. D22c and D22d),

$$\tau_E \langle \dot{z}_E \rangle = \tau_E v = U_{ES} e^{-(z_S - z_E)^2/8a^2} \langle z_S - z_E \rangle + \underbrace{U_{EF} e^{-(x - z_E)^2/8a^2} \langle x - z_E \rangle}_{\approx 0} \tag{E7}$$

$$\tau_S \langle \dot{z}_S \rangle = \tau_S v = U_{SE} e^{-(z_S - z_E)^2/8a^2} \langle z_E - z_S \rangle \tag{E8}$$

Combining Eq. (D22b) and Eq. (E8), we obtain the following relationship between velocity and separation between the bump positions,

$$\tau \left( U_{SE} e^{-(z_S - z_E)^2/8a^2} \right) v = U_{SE} e^{-(z_S - z_E)^2/8a^2} (z_E - z_S)$$

$$\Rightarrow \quad \tau v = z_E - z_S$$

which is the Eq. (16) in the main text.

To find how $v$ is related to the gain of SOM, we assume $(x - z_E)$ is still negligible,

$$\tau U_E v = U_{ES} e^{-\tau^2 v^2 / 8a^2}(-\tau v) \quad \Rightarrow \quad U_E = -U_{ES} e^{-\tau^2 v^2 / 8a^2}$$

Substituting the above equation into the height relation,

$$U_E = U_{EE} + U_{ES} e^{-\tau^2 v^2 / 8a^2} + U_{EF}$$

We arrive,

$$-2U_{ES} e^{-\tau^2 v^2 / 8a^2} = U_{EE} + U_{EF} \tag{E9}$$

From our previous simplifications, Eq. (D21), we know

$$U_{ES} = \frac{\rho a_S}{\sqrt{2} a_E} w_{ES} R_S$$

$$U_{ES} = \frac{\rho^2}{2} w_{ES} w_{SE} g_S R_E e^{-\tau^2 v^2 / 8a^2} \tag{E10}$$

which we can substitute back into Eq. (E9) to obtain,

$$-2 \cdot \frac{\rho^2}{2} w_{ES} w_{SE} g_S R_E e^{-\tau^2 v^2 / 4a^2} = \frac{\rho}{\sqrt{2}} w_{EE} R_E + \frac{\rho}{\sqrt{2}} w_{EF} R_F$$

$$\Leftrightarrow \left( \frac{\rho}{\sqrt{2}} w_{ES} w_{SE} g_S e^{-\tau^2 v^2 / 4a^2} + w_{EE} \right) R_E = -w_{EF} R_F.$$

Since $R_F$ is given, this means an increase $g_S$ comes with an increase of $v^2$ to keep the $R_F$ unchanged to satisfy the above equation. Solve the above equation,

$$v^2 = -\frac{4a^2}{\tau^2} \ln \left[ -\left( w_{EE} + \frac{w_{EF} R_F}{R_E} \right) \frac{\sqrt{2}}{\rho w_{ES} w_{SE}} \frac{1}{g_S} \right], \tag{E11}$$

$$= \frac{4a^2}{\tau^2} \left[ \ln g_S - \ln \left( \frac{\sqrt{2}(w_{EE} R_E + w_{EF} R_F)}{\rho(-w_{ES}) w_{SE} R_E} \right) \right], \tag{E12}$$

which becomes the Eq. (16) in the main text.

**The residue dynamics for circuit sampling**

The above analysis suggests the mean of the circuit samples, $\langle z_E \rangle$, captures the latent stimulus speed. Now we analyze the sampling dynamics of the residue that is defined as,

$$\delta z_E = z_E - \langle z_E \rangle, \quad \delta z_S = z_S - \langle z_S \rangle$$

and the residue of the input feature is similarly defined,

$$\delta x_t = x_t - \langle x_t \rangle.$$

Computing the difference between the circuit's bump position dynamics (Eqs. D22c and D22d) and the trial-averaged mean dynamics (Eqs. E7 and E8) yields the residue dynamics,

$$\tau_E \dot{\delta z_E} = U_{ES}(\delta z_S - \delta z_E) + U_{EF}(\delta x_t - \delta z_E) + \sigma_E \tau_E^{1/2} \xi_t, \tag{E13}$$

$$\tau_S \dot{\delta z_S} = U_{SE}(\delta z_E - \delta z_S). \tag{E14}$$

Considering the case that $(\delta z_S - \delta z_E)$ is small enough, which can be realized by a not strong SOM gain $g_S$, we can ignore it in $\delta z_E$ dynamics,

$$\tau_E \dot{\delta z_E} \approx U_{EF}(\delta x_t - \delta z_E) + \sigma_E \tau_E^{1/2} \xi_t.$$

We see the $\delta z_E$ dynamics is comparable to the circuit's Langevin sampling dynamics in the static case (Eq. E1). The above analysis has two implications. First, it suggests that the $\langle z_E \rangle$ captures the speed of the latent stimulus, which is generated from the separation of E and SOM's samples, i.e., $z_E - z_S$. Second, the residue dynamics $\delta z_E$ corresponds to a Langevin sampling dynamics to a latent stimulus with zero speed, in that $\langle \delta x_t \rangle = 0$.

### E.2 MIXED LANGEVIN/HAMILTONIAN SAMPLING

To ease of analysis, we convert the Eqs. (D23c and D23d) into the matrix form

$$\begin{pmatrix} \dot{z}_E \\ \dot{z}_S \end{pmatrix} = \mathbf{D}_U^{-1}\mathbf{F}_1 \begin{pmatrix} z_E \\ z_S \end{pmatrix} + \mathbf{D}_U^{-1}\boldsymbol{\mu}_z + \mathbf{D}_U^{-1/2}\Sigma_1\boldsymbol{\xi}_t \tag{E15}$$

where

$$\mathbf{D}_U = \begin{pmatrix} \tau U_E & \\ & \tau U_S \end{pmatrix}, \quad \mathbf{F}_1 = \begin{pmatrix} -(U_{EF} + U_{ES}) & U_{ES} \\ U_{SE} & -U_{SE} \end{pmatrix}, \quad \boldsymbol{\mu}_z = \begin{pmatrix} U_{EF}x_t \\ 0 \end{pmatrix}, \quad \Sigma_1 = \begin{pmatrix} \sigma_E & 0 \\ 0 & 0 \end{pmatrix} \tag{E16}$$

**Static mixed sampling ($x_t$ is fixed over time)**

To reveal how the circuit with SOM neurons implements Hamiltonian sampling, we can decompose the network dynamics $z_E$ as a mixture of the Langevin sampling and the Hamiltonian sampling parts Sale & Zhang (2024),

$$\tau_E\dot{z}_E = \underbrace{[U_{ES}(z_S - z_E) + U_{EF}^H(x_t - z_E)]}_{\text{Momentum } p, \text{ (Hamiltonian part)}} + \underbrace{[U_{EF}^L(x_t - z_E) + \sigma_E\sqrt{\tau_E}\xi_t]}_{\text{Langevin part}}, \tag{E17}$$

where $U_{EF}^H$ and $U_{EF}^L$ denotes the proportion of feedforward input contributed by the Hamiltonian or Langevin sampling component, respectively. From Eq. (E17), we define momentum $p$ as

$$p = U_{ES}(z_S - z_E) + U_{EF}^H(x - z_E)$$
$$= (-U_{ES} - U_{EF}^H, U_{ES}, U_{EF}^H) \cdot (z_E, z_S, x)^\top.$$

In this way, we can define a transition matrix between the network dynamics and momentum.

$$\begin{pmatrix} z \\ p \end{pmatrix} = \mathbf{T}\begin{pmatrix} z_E \\ z_S \end{pmatrix} + \begin{pmatrix} 0 \\ U_{EF}^H x_t \end{pmatrix}, \quad \mathbf{T} = \begin{pmatrix} 1 & 0 \\ -U_{ES} - U_{EF}^H & U_{ES} \end{pmatrix}.$$

To simplify the analysis, we consider $x_t = 0$ over time without loss of generality (and then $\dot{x}_t = 0$:

$$\frac{d}{dt}\begin{pmatrix} z \\ p \end{pmatrix} = \mathbf{T}\frac{d}{dt}\begin{pmatrix} z_E \\ z_S \end{pmatrix} = (\mathbf{T}\mathbf{D}_U^{-1}\mathbf{M}_U\mathbf{T}^{-1})\underbrace{\mathbf{T}\begin{pmatrix} z_E \\ z_S \end{pmatrix}}_{(z,p)^T} + \mathbf{T}\boldsymbol{\mu}_z + \mathbf{T}\mathbf{D}_U^{-1/2}\Sigma\,\xi_t$$

To derive $\mathbf{T}\mathbf{D}_U^{-1}\mathbf{M}_U\mathbf{T}^{-1}$,

$$\mathbf{T}^{-1} = \frac{1}{U_{ES}}\begin{pmatrix} U_{ES} & 0 \\ U_{ES} + U_{EF}^H & 1 \end{pmatrix}$$

$$\mathbf{D}_U^{-1}\mathbf{M}_U = \begin{pmatrix} \tau_E^{-1} & 0 \\ 0 & \tau_S^{-1} \end{pmatrix}\begin{pmatrix} -(U_{EF} + U_{ES}) & U_{ES} \\ U_{SE} & -U_{SE} \end{pmatrix} = \begin{pmatrix} -\tau_E^{-1}(U_{EF} + U_{ES}) & \tau_E^{-1}U_{ES} \\ \tau_S^{-1}U_{SE} & -\tau_S^{-1}U_{SE} \end{pmatrix}$$

Then, multiply and simplify to obtain the result,

$$\mathbf{T}\mathbf{D}_U^{-1}\mathbf{M}_U\mathbf{T}^{-1} = \frac{1}{U_{ES}}\begin{pmatrix} -\tau_E^{-1}U_{EF}^L & \tau_E^{-1} \\ U_{EF}^L h_E - \tau_S^{-1}U_{SE}U_{EF}^H & -h_E - \tau_S^{-1}U_{SE} \end{pmatrix}$$

where $h_E = \tau_E^{-1}(U_{ES} + U_{EF}^H)$.

We then rewrite the dynamics as,

$$\frac{d}{dt}\begin{pmatrix} z \\ p \end{pmatrix} = -\begin{pmatrix} \tau_E^{-1}U_{EF}^L & -\tau_E^{-1} \\ \beta_E & \beta_p \end{pmatrix}\begin{pmatrix} z \\ p \end{pmatrix} + \mathbf{T}\boldsymbol{\mu}_z + \begin{pmatrix} \tau_E^{-1/2}\sigma_E \\ \sigma_p \end{pmatrix}\boldsymbol{\xi}_t \tag{E18}$$

where

$$\beta_E = -\tau_E^{-1}U_{EF}^L(U_{ES} + U_{EF}^H) + \tau_S^{-1}U_{SE}U_{EF}^H \tag{E19}$$
$$\beta_p = \tau_E^{-1}(U_{ES} + U_{EF}^H) + \tau_S^{-1}U_{SE} \tag{E20}$$
$$\sigma_p^2 = (U_{ES} + U_{EF}^H)^2\sigma_E^2\tau_E^{-1} \tag{E21}$$

**Mapping to the standard form of mixed Langevin and Hamiltonian sampling**

In mixed sampling, the equilibrium distribution $\pi(z)$ sampled is defined as,

$$\pi(z,p) = \exp[-H(z,p)] = \exp[\ln \pi(z) - K(p)] \tag{E22}$$

where $K(p)$ is kinetic energy with $m$ analogous to the mass in physics. As in the main text, the Hamiltonian sampling dynamics with friction $\gamma$ for dampening momentum is Chen et al. (2014); Ma et al. (2015),

$$\frac{d}{dt} \begin{bmatrix} \tilde{z}_t \\ p_t \end{bmatrix} = - \begin{bmatrix} 0 & -\tau_H \\ \tau_H & \gamma \end{bmatrix} \begin{bmatrix} -\nabla_z \ln \pi(z) \\ m^{-1}p \end{bmatrix} + \sqrt{2} \begin{bmatrix} 0 & 0 \\ 0 & \gamma^{1/2} \end{bmatrix} \boldsymbol{\xi}_t. \tag{E23}$$

For mixed sampling Langevin and Hamiltonian sampling,

$$\frac{d}{dt} \begin{pmatrix} z \\ p \end{pmatrix} = - \begin{pmatrix} \tau_L^{-1} & -\tau_H^{-1} \\ \tau_H^{-1} & \tau_p^{-1} \end{pmatrix} \begin{pmatrix} -\nabla_z \ln \pi(z) \\ m^{-1}p \end{pmatrix} + \sqrt{2} \begin{pmatrix} \tau_L & \\ & \tau_p \end{pmatrix}^{-1/2} \eta_t \tag{E24}$$

where $\nabla_z \ln \pi(z) = \Lambda_F(x_t - z) = -\Lambda_F z$ (considering $x_t = 0$).

We then convert the circuit's bump position dynamics (Eq. E18) into the standard mixed sampling form,

$$\frac{d}{dt} \begin{pmatrix} z \\ p \end{pmatrix} = - \begin{pmatrix} U_{EF}^L(\tau_E\Lambda_F)^{-1} & -\beta_E\Lambda_F^{-1} \\ \beta_E\Lambda_F^{-1} & \tau_E\beta_p\beta_E\Lambda_F^{-1} \end{pmatrix} \begin{pmatrix} \Lambda_F z \\ (\tau_E\beta_E)^{-1}\Lambda_F p \end{pmatrix} + \begin{pmatrix} \tau_E^{-1/2}\sigma_E \\ \sigma_p \end{pmatrix} \eta_t \tag{E25}$$

Comparing Eq. (E24) to Eq. (E18), we have,

$$\tau_L^{-1} = U_{EF}^L(\tau_E\Lambda_F)^{-1}, \tag{E26a}$$

$$\tau_H^{-1} = \beta_E\Lambda_F^{-1}, \tag{E26b}$$

$$m^{-1} = (\tau_E\beta_E)^{-1}\Lambda_F^{-1} \tag{E26c}$$

$$\tau_p^{-1} = \beta_p m^{-1} \rightarrow \tau_p^{-1} = \tau_E\beta_p\beta_E\Lambda_F^{-1} \tag{E26d}$$

We next determine the conditions in which the network can realize mixed Langevin and Hamiltonian sampling. For the Langevin sampling condition:

$$\tau_L^{-1} = U_{EF}^L(\tau_E\Lambda_F)^{-1} = \frac{\tau_E^{-1}\sigma_E^2}{2}, \tag{E27}$$

where,

$$U_{EF} = \frac{\rho w_{EF}}{\sqrt{2}}, \quad R_F = \frac{a}{\sqrt{2\pi}}\Lambda_F,$$

We can then constrain the feedforward weight for realizing Langevin sampling component,

$$w_{EF}^L = \frac{\sqrt{\pi}\sigma_E^2}{U_{EF}^L a} = \left(\frac{2}{\sqrt{3}}\right)^3 \frac{\mathsf{F}}{U_{EF}^L} \tag{E28}$$

In addition, realizing the Hamiltonian sampling in the circuit requires,

$$\tau_E\beta_p\beta_E\Lambda_F^{-1} = \frac{\sigma_p^2}{2}. \tag{E29}$$

Substituting the expressions (Eq. E21) into the above equation, and define common terms to simplify the expression,

$$h_E = \tau_E^{-1}(U_{ES} + U_{EF}^H), \quad h_S = \tau_S^{-1}U_{SE} \tag{E30}$$

We arrive,

$$(h_E + h_S)(-U_{EF}^L h_E + U_{EF}^H h_S)\Lambda_F^{-1} = U_{EF}^L h_E^2 \tag{E31}$$

Rearranging the above equation into a quadratic for $h_E$,

$$2U_{EF}^L h_E^2 + (U_{EF}^L - U_{EF}^H)h_S h_E - U_{EF}^H h_S^2 = 0 \tag{E32}$$

By using the relation that $U_{EF}^H/U_{EF}^L = w_{EF}^H/w_{EF}^L$, the root of $h_E$ is,

$$h_E = \frac{h_S}{4}\left[-\left(1 - \frac{w_{EF}^H}{w_{EF}^L}\right) \pm \sqrt{\left(1 - \frac{w_{EF}^H}{w_{EF}^L}\right)^2 + 8\frac{w_{EF}^H}{w_{EF}^L}}\right] \equiv F(w_{EF}^H/w_{EF}^L) \cdot h_S. \qquad \text{(E33)}$$

Combining the above equation with the Eq. (E30)

$$\tau_E^{-1}(U_{ES} + U_{EF}^H) = F(w_{EF}^H/w_{EF}^L)\tau_S^{-1}U_{SE} \qquad \text{(E34)}$$

Substituting the detailed expression of $U_{ES}$, $U_{SE}$, $\tau_E$, we can find

$$\left(U_S w_{ES}\right) \cdot g_S - R_F \cdot w_{EF}^H = F\left(w_{EF}^H/w_{EF}^L\right)U_E, \qquad \text{(E35)}$$

which is the Eq. (19) in the main text.

Note that the $w_{EF}^H$ is the extra feedforward weight for Hamiltonian sampling based on the original weight $w_{EF}^L$ for Langevin sampling. And the extra $w_{EF}^H$ needs to be associated with the SOM gain $g_S$. We see that there is a line manifold of the combination of $w_{EF}^H$ and $g_S$ implements the Hamiltonian sampling in the circuit.

Dynamic mixed sampling (non-zero speed, noisy latent transition) We present how the circuit can realize the mixed Langevin and Hamiltonian sequential sampling to implement a latent stimulus with non-zero speed and noisy transitions over time.

The overall math analysis process is similar to the Langevin sequential sampling as presented in Sec. E.1.1. That is, we need the SOM gain to enable the internal speed generation in the circuit that captures the latent stimulus speed (Eq. E7), and then the residue dynamics is equivalent to implementing the sampling of a static input with only noisy transitions (Eqs. E13 - E14).

Unlike the Langevin sequential sampling where the residue $\delta z_E - \delta z_S$ is negligible (Eq. E15) in that all the SOM gain $g_S$ is used to generate internal speed, in the Hamiltonian sequential sampling the $\delta z_E - \delta z_S$ is non-negligible. Therefore, we need to analyze the joint residue dynamics of $\delta z_E$ and $\delta z_S$. Copy the Eqs. (E7-E8 and E13 - E14) in below,

$$\tau_E \langle \dot{z}_E \rangle \approx U_{ES}\langle z_S - z_E \rangle, \qquad \text{(E36)}$$

$$\tau_S \langle \dot{z}_S \rangle = U_{SE}\langle z_E - z_S \rangle, \qquad \text{(E37)}$$

$$\tau_E \dot{\delta z_E} = U_{ES}^H(\delta z_S - \delta z_E) + U_{EF}^H(\delta x_t - \delta z_E) + \sigma_E \tau_E^{1/2}\xi_t, \qquad \text{(E38)}$$

$$\tau_S \dot{\delta z_S} = U_{SE}^H(\delta z_E - \delta z_S). \qquad \text{(E39)}$$

We can think of the $\langle z_X \rangle$ and $\langle \delta z_X \rangle$ ($X = E, S$) in two steps.

1. First, we determine the circuit weight to make sure $\langle z_X \rangle$ tracks the input speed, including setting the speed-dependent SOM gain by using Eq. (E12) and the feedforward weight by using Eq. E4.

2. Second, based on the circuit weight in the first step, we overlay additional feedforward input $U_{EF}^H$ and additional SOM inhibition $U_{ES}^H$ to induce oscillations in the residue $\langle \delta z_X \rangle$ dynamics. In this way, the residue dynamics obey the same math analysis with Sec. E.2. This immediately gives rise to the additional feedforward weight and SOM gain to realize Hamiltonian sequential sampling,

$$\left(U_S w_{ES}\right) \cdot g_S^H - R_F \cdot w_{EF}^H = F\left(w_{EF}^H/w_{EF}^L\right)U_E, \qquad \text{(E40)}$$

which is the Eq. (19) in the main text.

E.3 HIGH-DIMENSIONAL POSTERIOR DISTRIBUTIONS

We consider the multivariate posterior distribution case via coupled circuits (See Supp Fig S6). The core algorithm (Eq. 9) generalizes naturally to multivariate latent states, where the transition probability becomes a joint distribution and the feedforward input shapes a multivariate likelihood.

$$\pi_{t+1}(z_{1,t+1}, z_{2,t+1}) \propto f(z_{1,t+1}, z_{2,t+1}) \cdot \left[\frac{1}{L}\sum_{l=1}^{L} p\left(z_{1,t+1}, z_{2,t+1}|\tilde{z}_{1,t}^{(l)}, \tilde{z}_{2,t}^{(l)}\right)\right] \qquad \text{(E41)}$$

where $z_1$ and $z_2$ are the two latent stimuli. Each latent stimulus can be sampled by a recurrent circuit motif that is the same as Fig. 1B, while the two circuit motifs are coupled together with their coupling storing the prior $p(z_1, z_2)$. When supposing only one sample is generated in each time step, the instantaneous posteriors is approximated as,

$$\pi_{t+1}(z_{1,t+1}, z_{2,t+1}) \approx f(z_{1,t+1}, z_{2,t+1}) p\big(z_{1,t+1}, z_{2,t+1} | \tilde{z}_{1,t}^{(l)}, \tilde{z}_{2,t}^{(l)}\big) \tag{E42}$$

In particular, when assuming each input is independently generated by the latent stimulus, the likelihood can be factorized, i.e., $f(z_{1,t+1}, z_{2,t+1}) = f(z_{1,t+1}) f(z_{2,t+1})$. Similarly, we consider the transition probability can be factorized, i.e., $p\big(z_{1,t+1}, z_{2,t+1} | \tilde{z}_{1,t}^{(l)}, \tilde{z}_{2,t}^{(l)}\big) = p\big(z_{1,t+1} | \tilde{z}_{1,t}^{(l)}\big) p\big(z_{1,t+1} | \tilde{z}_{2,t}^{(l)}\big)$, the marginal instantaneous posterior of $z_1$ is,

$$\pi_{t+1}(z_{1,t+1}) \approx f(z_{1,t+1}) p\big(z_{1,t+1} | \tilde{z}_{1,t}^{(l)}\big) p\big(z_{1,t+1} | \tilde{z}_{2,t}^{(l)}\big) \tag{E43}$$

Then we can plug this expression into Eq. (11), and then the dynamics is consistent with the dynamics of the circuit motif 1 in the coupled motifs. And the terms on the RHS of the above equation corresponds to the feedforward input, recurrent input within the same circuit motif, and the recurrent input from another circuit input.

### E.4 BIAS VARIANCE TRADE-OFF

The sampling time constant governs the bias-variance trade-off. Below we analyze the equilibrium mean and variance of the sampling error for Langevin sequential sampling. The Langevin sequential sampling has the following sampling dynamics (Eq. 11),

$$\tilde{z}_t = \tilde{z}_{t-1} + (\tau_L^{-1}\delta t)\nabla_z \ln \pi_t(\tilde{z}_{t-1}) + (2\tau_L^{-1}\delta t)^{1/2}\eta_{t-1} \tag{E44}$$

where $\eta_t \sim \mathcal{N}(0, I)$. From the Eq. (10), the gradient is given by

$$\nabla_z \ln \pi_t(\tilde{z}_{t-1}) = \Omega_t(\mu_t - \tilde{z}_{t-1}) = \Lambda_F(x_t - \tilde{z}_{t-1}) + \Lambda_z v$$

Then,

$$\tilde{z}_t = \tilde{z}_{t-1} + (\tau_L^{-1}\delta t)[\Lambda_F(x_t - \tilde{z}_{t-1}) + \Lambda_z v] + (2\tau_L^{-1}\delta t)^{1/2}\eta_t \tag{E45}$$

Here our analysis assumes the sampler's internal speed ($v$ in Eq. (E45)) matches the true speed of the latent stimulus in the external world (Eq. 6). This corresponds to set the SOM inhibition in the recurrent circuit model to make the circuit's internal speed matches the true speed (Eq. 20)

From Eq. 7, the observed feature $x_t$ is generated by,

$$x_t = z_t + \Lambda_F^{-1/2}\zeta_t \tag{E46}$$

where $z_t$ is the true latent stimulus. Substituting the above equation of $x_t$ into Eq. ((E45)),

$$\tilde{z}_t = \tilde{z}_{t-1} + (\tau_L^{-1}\delta t)[\Lambda_F(z_t + \Lambda_F^{-1/2}\zeta_t - \tilde{z}_{t-1}) + \Lambda_z v] + (2\tau_L^{-1}\delta t)^{1/2}\eta_{t-1} \tag{E47}$$

Meanwhile, the dynamics of the true latent stimulus is derived from the transition probability (Eq. 6)

$$z_t = z_{t-1} + v\delta t + \Lambda_z^{-1/2}\sqrt{\delta t}\xi_{t-1} \tag{E48}$$

Define the error between the sample and the true latent stimulus as

$$e_t = \tilde{z}_t - z_t \tag{E49}$$

And subtracting both sides of Eq. ((E47)) by the both sides of Eq. ((E48)) respectively,

$$e_t = e_{t-1} + (\tau_L^{-1}\delta t)[\Lambda_F(z_t + \Lambda_F^{-1/2}\zeta_t - \tilde{z}_{t-1}) + \Lambda_z v] + (2\tau_L^{-1}\delta t)^{1/2}\eta_{t-1} - [v\delta t + \Lambda_z^{-1/2}\sqrt{\delta t}\xi_{t-1}] \tag{E50}$$

Meanwhile, the $z_t - \tilde{z}_{t-1}$ in the 2nd RHS term in Eq. (E50) can be calculated

$$z_t - \tilde{z}_{t-1} = (z_{t-1} - \tilde{z}_{t-1}) + v\delta t + \Lambda_z^{-1/2}\sqrt{\delta t}\xi_{t-1} = -e_{t-1} + v\delta t + \Lambda_z^{-1/2}\sqrt{\delta t}\xi_{t-1} \tag{E51}$$

Substituting this back into Eq. (E50), and reorganize the equation,

$$\frac{e_t - e_{t-1}}{\delta t} = -\tau_L^{-1}\Lambda_F e_{t-1} + (\Lambda_F \tau_L^{-1}\delta t + \tau_L^{-1}\Lambda_z - 1)v + noise_t \tag{E52}$$

where the noise term is

$$noise_t = (\Lambda_F \tau_L^{-1} \delta t - 1)\Lambda_z^{-1/2}\xi_{t-1} + \tau_L^{-1}\Lambda_F^{1/2}\zeta_t + (2\tau_L^{-1})^{1/2}\eta_{t-1} \tag{E53}$$

Then we convert the difference equation of $e_t$ into a differential equation by taking to the limit $\delta t \to 0$. Note that the terms containing $\delta t$ on the RHS disappear as they are high-order terms,

$$\lim_{\delta t \to 0} \frac{e_t - e_{t-1}}{\delta t} = \frac{de_t}{dt} = -\tau_L^{-1}\Lambda_F e_{t-1} + (\tau_L^{-1}\Lambda_z - 1)v + [-\Lambda_z^{-1/2}\xi_{t-1} + \tau_L^{-1}\Lambda_F^{1/2}\zeta_t + (2\tau_L^{-1})^{1/2}\eta_{t-1}] \tag{E54}$$

**Sampling Bias** To eliminate the bias, i.e., $\langle e_t \rangle = 0$, we need to let the drift bias term (the 2nd RHS term in the above equation) be zero, corresponding

$$\tau_L = \Lambda_z \tag{E55}$$

So it exists an optimal sampling time constant.

**Sampling variance** The sampling error variance in the equilibrium can be immediately solved by using the Lyapunov equation (note that the three noises are independent),

$$V(e_t) = \frac{\Lambda_z^{-1} + \tau_L^{-2}\Lambda_F + 2\tau_L^{-1}}{2\tau_L^{-1}\Lambda_F} \tag{E56}$$

# F    SIMULATION DETAILS

## F.1    NETWORK PARAMETERS AND SIMULATION

The commonly used parameters in all simulations are included in Table 1. Each network, both excitatory and inhibitory, includes $N = 180$ neurons, uniformly distributed along the stimulus feature space $z \in (-180°, 180°]$. The neuronal density is $\rho = N/w_z$ where $w_z = 360$ is the width for stimulus feature space.

The synaptic weights are scaled by the minimal E-to-E recurrent, $w_c$ connection needed to hold persistent activity without feedforward input or SOM gain, solved by when setting $R_F = g_S = 0$.

$$w_c = 2\sqrt{2}(2\pi)^{1/4}\sqrt{ka_E/\rho} \approx 0.896. \tag{F57}$$

We then scale the feedforward input intensity by the peak population synaptic input, $U_c$, calculated as,

$$U_c = \frac{w_c}{2\sqrt{\pi}ka_E} \tag{F58}$$

Table 1: Default network parameters

| Parameter | Variable | Value |
|---|---|---|
| Excitatory time constant | $\tau$ | 1 |
| Feedforward weight | $w_{EF}$ | $0.83w_c$ |
| E to SOM weight | $w_{SE}$ | $0.5w_c$ |
| PV to E weight | $w_{PV}$ | 0.0005 |
| SOM to E weight | $w_{ES}$ | $0.5w_c$ |
| Connection width | $a_E$ | 40° |
| Feedforward input location | $z$ | 0 |
| Fano factor of injected variability | F | 0.5 |
| SOM connection width | $a_S$ | 37.4° |
| E to SOM connection width | $a_{SE}$ | 34.6° |
| SOM to E connection width | $a_{ES}$ | 20° |
| SOM Time constant | $\tau$ | $5\tau$ |

When $\Lambda_z > 0, v = 0$ in Fig. 2 C-D, I, the feedforward input intensity is, $R_F = 0.2U_c$. Afterwards, the recurrent weight, standard deviation for the transition probability, and feedforward input intensity

are set based on the parameter scan in 2D, to $w_{EE} = 0.53w_c$, $R_F = 0.8U_c$, and $\Lambda_z^{-1/2} = 0.04$, respectively for all following simulations.

Simulations of the network dynamics were done using Euler's method. The time step was $dt = 0.01\tau$. Each stimulation was run for $500\tau$ with the first $100\tau$ discarded to exclude non-equilibrium responses. Each simulation took approximately one minute on a Asus ROG Zephyrus laptop which has an i7 intel core and 32 RAM. For parameter scans, a HPC 512 GB RAM computing cluster was utilized with 36 parallel jobs for about 5 minutes.

Table 2: Langevin network parameters Fig. 2

| Parameter | Variable | Value |
|---|---|---|
| Feedforward weight | $w_{EF}$ | $0.8w_c$ |
| Feedforward input location | $z$ | $0$ |
| External speed | $v$ | $0.83$ |
| SOM gain | $g_S$ | $1.25$ |
| Transition standard deviation | $\sigma_z = \Lambda_z^{-1/2}$ | $0.04$ |
| Feedforward input intensity | $R_F$ | $0.8U_c$ |

Table 3: Hamiltonian network parameters

| Parameter | Variable | Value |
|---|---|---|
| Feedforward weight | $w_{EF}$ | $1.53w_c$ |
| E to E weight (Fig. 3C-F) | $w_{EE}$ | $1.3w_c$ |
| Feedforward input location | $z$ | $0$ |
| External speed | $v$ | $2.5$ |
| SOM gain (Fig. 3C-F) | $g_S$ | $10$ |

Table 4: Fig. A4 network parameters

| Parameter | Variable | Value |
|---|---|---|
| Feedforward weight | $w_{EF}$ | $0.83w_c$ |
| E to SOM weight | $w_{SE}$ | $0.5w_c$ |
| E to E weight | $w_{EE}$ | $0.5w_c$ |
| SOM to E weight | $w_{ES}$ | $0.5w_c$ |
| SOM gain | $g_S$ | $5$ |
| E1 to E2 weight | $w_{12} = w_{21}$ | $0.2w_c$ |

F.2 READ OUT STIMULUS SAMPLES FROM THE POPULATION RESPONSES

Instantaneous stimulus samples, $z_E, z_S$ were read out with a linear decoder, population vector from the neuron population.

$$z_E(t) = \frac{\sum_j \mathbf{r}_E(\theta_j, t)\theta_j}{\sum_j \mathbf{r}_E(\theta_j, t)} \tag{F59}$$

The empirical distribution of the stimulus samples was defined as,

$$p(z) = \sum_t \delta(z - z_E(t)) \tag{F60}$$

F.3 COMPARING THE SAMPLING DISTRIBUTIONS WITH POSTERIORS

The Kullback-Leibler divergence was used as an metric for the difference between the sampling distribution, $p(z) = \sum_t \delta(z - z_E(t))$, and theoretically calculated posterior distribution, $p(z|\mathbf{r}_F)$.

$$D_{KL}[p(z|\mathbf{r}_F)||p(z)] = \int p(z|\mathbf{r}_F) \ln \frac{p(z|\mathbf{r}_F)}{p(z)} dz \tag{F61}$$

The posterior, or likelihood since the prior is uniform, is read out from the feedforward input Eq.6. We parameterized the empirical sampling distribution as a Gaussian to use the mean and covariance for the samples to calculate the KL Divergence.

### F.4 REPRODUCING E NEURONS' TUNING CURVES FROM MODULATING INTERNEURONS

For comparison of the E neurons' tuning curves, Wilson et al. (2012), we perturb PV and SOM neurons' in the network individually and measure how these perturbations change E neurons' tuning curves. The experiments applied a full-field light to the same type of neuron Wilson et al. (2012), which we approximate as a constant input applied to each neuron of the same type.

For SOM,

$$\tau \frac{\partial \mathbf{u}_S(x,t)}{\partial t} = -\mathbf{u}_S(x,t) + \rho \mathbf{W}_{SE} * \mathbf{r}_E(\theta,t) + I_S; \tag{F62}$$

where $I_S$ is the additional input applied to each SOM neuron.

Similarly for the PV neurons into the divisive normalization function, (Eq. 1b),

$$\mathbf{r}_E(\theta,t) = \frac{[\mathbf{u}_E(\theta,t)]_+^2}{1 + \rho w_{EP} \int ([\mathbf{u}_E(\theta,t)]_+^2 + I_P) d\theta'}, \tag{F63}$$

where $I_P$ is the perturbing input.

Then, with the existence of one of these offset inputs, we change the presented feedforward input location $z$ (Eq. 1e) and measure the mean firing rate of an example E neuron.

### F.4.1 CONTINUOUS APPROXIMATION OF THE POISSON FEEDFORWARD INPUTS

In modeling the sensory input to the network, we approximate Poisson variability by a Gaussian distribution, a standard approach when firing rates are sufficiently high. Specifically, the feedforward input $\mathbf{r}_F(\theta,t)$ is stochastically evoked from the latent stimulus $z_t$, with a mean firing rate $\lambda_F(\theta|z_t)$ given by a Gaussian tuning curve. Under a Gaussian approximation to the Poisson process, $\mathbf{r}_F(\theta,t)$ is treated as a continuous random variable:

$$\tilde{r}_F(\theta,t) = \lambda_F(\theta|z_t) + \sqrt{\lambda_F(\theta|z_t)}, \xi(\theta,t), \tag{F64}$$

where $\xi(\theta,t)$ denotes independent standard Gaussian noise. In the Hidden Markov Model (HMM) framework, two sources of stochasticity are naturally present: one from the internal latent dynamics of $z_t$, which evolves with its own noise process, and another from the observations $\mathbf{r}_F(\theta,t)$, which reflects noisy sensory encoding of the latent state.

