# OpenReview forum: "Canonical cortical circuits: A unified sampling machine for static and dynamic inference"
_ICLR.cc/2026/Conference — ICLR 2026 Conference Withdrawn Submission_

### Official Review · Reviewer_7SmZ · 2025-10-28

**Soundness:** 2
**Presentation:** 3
**Contribution:** 2
**Rating:** 2
**Confidence:** 5

**Summary:**

The paper adapts a bump attractor network with two kinds of inhibition and intrinsic receiving stimulus-tuned poisson input (originally developed in Sale&Zhang2024 and preceding); it uses it as a way to implement different forms of sampling from a distribution over a one dimensional circular variable.  Mechanistically, the attractor dynamics make for a stable bump of activity which is interpreted as a parametric

The dynamic inference takes the form of linear gaussian dynamics whose solution is equivalent to a Kalman filter.

**Strengths:**

The recurrent circuit matches coarse anatomy of cortical circuitry and can be mathematically traced back to long history of mathematically tractable attractor dynamics.

The parametric nature of the encoding allows for the circuit to process inputs with varying degrees of uncertainty without having to adjust synaptic strengths, which is particularly interest for dynamic stimuli  (this was true about old versions of the model for static inference but is also inherited to this variant).

Although the text is dense in places, it is generally clear what was done and how (bar the explanation of the mapping between sampling dynamics and circuit elements which could need clarification and additional justification).

**Weaknesses:**

The conceptualization of the problem as a whole: The general logic of using sampling-based inference for computations that are fundamentally parametric using a network that explicitly encodes the parametric form with sampling on top lacks the even the most basic normative justification. It's sampling for sampling's sake rather than in service of actually achieving representational gains for complex high-dimensional posteriors, which was the original logic of neural sampling as introduced by Hoyer & Hyvärinen, 2003 and Fiser et al 2010 and the way it has been used to relate to visual neural activity in later works like those from Orban and Echeveste. Computationally, I fail to see the point of why do things this way. A lot of the logic feels backward to me as it forces an assumption of world statistics needing to match the constraint of the circuit dynamics rather than the typical other way around.

Computational limits: The going from well established attractor dynamics to an induced model is also extremely restrictive and allows little to no variation outside of a very simple generative model form. This means the ability of such a model to generalize across natural world statistics in different modalities or even outside the simple low-d linear gaussian set difficult if not impossible.

Validation of the quality of sampling: given the very coarse approximation which replaces the full prior form one step back with a single sample from it, it is not clear neither mathematically nor numerically that the resulting system is actually sampling from the same posterior density as the ground truth kalman filter. at the very minimum the quality of the approximation is expected to decrease with posterior or sensory uncertainty which i have not seen analyzed in any systematic way in the text but seems critical for demonstrating the circuit actually achieves the claimed computational function.

Comparison to previous model alternatives: i would have liked to see a more precise discussion about predictions that the model makes which are actually distinguishable from previous proposals. usually with sampling-based representations the mean responses do not distinguish form a parametric solution such as that from Deneve and Pouget 2007 (although in this instance the fact that the alternative does exact close to exact kalman whereas this relies on a coarse approximation for the propagation of uncertainty over time may already have signatures in readout behavior or neural mean responses, not sure); distinguishable features in this space would usually take the form of predictions about response variability but i did not see anything of the form in the text.

Novelty and significance: at the technical level this seems like an incremental variation of the previous models in Zhang et al, Sale et al, with complete equivalence for the static case and minimal change for the kalman addition.

**Questions:**

Description of dynamics in Eq 1a-e is missing details. Can you please make sure you list and explain each variable appearing there.

What is the computational purpose of Hamiltonian sampling for a 1d density? The use of Hamiltonian dynamics (Aitchison), or nonnormal dynamics (Hennequin) or other forms of annealed sampling (Savin) have historically been invoked in the service of sampling fast from complex distributions which are either heavily correlated or multimodal. I fail to see the logic of doing the same here. Please explain.

Missing literature: several Kalman filtering implementations have been proposed in the lit. that do not seem referred to, e.g. Deneve et al 2007 "Optimal Sensorimotor Integration in Recurrent Cortical Networks: A Neural Implementation of Kalman Filters" which do not require the very coarse computational approximations due 1 sample per step updates.

The math in the sampling section makes some implicit assumptions w.r.t. the map between the inferred latent variable z_t and the population responses of the neurons in the circuit u or r_e. Please clarify this map explicitly in the main text. More generally, the exact mapping between the neural dynamics elements and the dynamic inference implementation could be generally more explicit and better explained.

Can you comment on the biophysical nature of the mechanism that controls the gain of the SOM subpopulation and how is that supposed to be calibrated to the true speed of the moving sensory stimulus?

the faster sampling with faster speed is misleading as a mathematical statement" sampling speed refers to the autocorrelation of MCMC samples drawn from the same density whereas here we are talking about changes across samples driven by temporal changes in the mean of the density itself. this seems formally incorrect as a statement.

Hamiltonian dynamics: can you please comment on the exact map between the variables of the sampler and the neural activity, in particular how is the momentum variable encoded in neural activity? that was not clear in either the static nor the dynamic case. The way i understand the approach as a whole it is a matter of establishing a direct map between dynamic variables and then work out by structural equation identification the expression for various pieces of the connectivity so i don't quite get how one can do that without having some set of neural variables explicitly implementing the momentum auxiliary variable. Or if there are such variables how do they fit into the canonical circuit definition presented at the start.

Is the momentum variable also estimated via single samples? how do you ensure that the dynamics as a whole remain volume preserving which such a coarse approximation ? what guarantees that the resulting circuit samples from the right distribution and does not accummulate errors over time? is there a formal proof for that statement or something you observe empirically and if the latter then under what kind of conditions?

What is the size of the networks numerically simulated in Fig2?
is this the prediction of the asymptotic limit or the dynamics of a finite simulated network?
Same question for Fig3

What makes the oscillations of hamiltonian dynamics (well documented in past work including Aitchison) have anything to do specifically with hippocampal activity? how is a simple kalman filter in 1d related to hippocampal computations? can you please expand on the nature of the analogy you are trying to build between model and data and at what qualitative level should one evaluate that comparison
What is unique about it to your model as opposed to more explicit internal models of hippocampus based on sampling proposed before, e.g. by Ujfalussy and Orban?

Discussion: based on the textbook definition of the term HMMs assume discrete latent variables whereas the speed of stimulus is continuous, why talk about alpha-beta type of inference and smoothing in HMMs specifically when 1) filtering is all you do for kalman inference and 2) a hierarchical kalman seems much more suited to the continuous nature of the latent variable in question. Please explain or correct.

---

> ### Author Response · Authors · 2025-11-21
>
> Thanks very much for the reviewer’s positive comments on our clear writing, and the circuit can track dynamic input with varying uncertainties. Below we reply to your weaknesses and questions one by one.
>
> ## Weaknesses
>
> ### 1. Conceptual framework
>
> We appreciate the reviewer raising an important question about the significance of our study, which we didn’t emphasize sufficiently well due to the page limit. Apart from gaining representational flexibility, one implication from our study is the sampling can also simplify the complexity of circuit implementation, which is an important issue in studying circuit algorithms while has not been paid much attention before. Please refer to the section “Sampling simplifies circuit implementation” in Global reply. We also emphasize this new significance of sampling in the revised manuscript with an extra page (lines 497-507).
>
> ### 2. Computational limits
>
> We thank the reviewer for their comments and questions.  Since this is a common concern, we address it in the global reply. Please refer to the section “Complex generative models” there.
>
>
> ### 3. Sampling validation
>
> Thank you for allowing us to clarify.  In Eq. 7-8, the predictive posterior distribution (the integration term) is actually a __hybrid__ representation, rather than a pure sampling-based representation from a single sample that collapses into a delta function. Specifically, the predictive posterior in Eq. 7 is approximated by the distribution $p(z_{t+1}|\tilde{z}_t)$ in Eq. 8, which is a parametric Gaussian distribution that maps to the parametric recurrent E interactions, but its mean parameter depends on the sample $\tilde{z}_t$.
> We expand the text after Eq. 8 to avoid this confusion in the revised manuscript in section 3.2
> We did analytically perform the error analysis in Appendix E4 . The dense text may lead readers easily ignore that. Our analysis shows a closed form expression for the sampling bias and variance.  Specifically, the sampling bias can be eliminated when the sampling time constant satisfies $\tau_L=\Lambda_z$ (Eq. E55). And the sampling variance (Eq. E56) does increases with higher uncertainty (e.g., smaller $\Lambda_F$).   In simulation, we use MSE to quantify the sampling error. And the relation between sampling error and uncertainty can be seen from Fig. 2I, bottom panel, shaded region.
>
>
> ### 4. Model comparison
>
> This is a superb question especially in the context of the long-term debate between parametric and sampling circuits. The difference between Kalman and Hamiltonian sequential sampling is obvious: the latter introduces oscillations around a moving latent stimulus, which is a feature absent in Kalman. However, distinguishing the stimulus estimates made from Kalman and the Langevin sequential sampling is tricky in practice, because the former looks like a noise-free version of the latter as you said. And if we remove the internal circuit variability (Eq. 1a; last term), our circuit without SOM looks basically the same as Deneve 07. In principle, Kalman and Langevin can be distinguished by the variability: if we _clamp_ the instantaneous posterior ($\pi_t(z_t)$ in Eq. 8) and run the dynamics for multiple trials, the Langevin circuit will exhibit variability whose variance is determined by $\pi_t(z_t)$, whereas as the Kalman circuit won’t have this variability. However, this comparison is probably implausible in practical data analysis, because we cannot clamp the instantaneous posterior, and the amount of neural data might not be sufficient to have the statistical power.
> We added a new paragraph “Distinguish deterministic and sampling circuits in dynamic inference” in the Discussion in the revised manuscript (line 512).
>
> ### 5. Novelty and significance
>
> Thank you for your detailed reply.  Since this feedback is a common concern, we address this in the global Official Comments. Please refer to the section “Novelty” there.

---

> ### Author Response · Authors · 2025-11-21
>
> ## Questions
>
> > 1. Description of dynamics in Eq 1a-e is missing details. Can you please make sure you list and explain each variable appearing there.
>
> Due to the limited space, we compressed the explanation of the circuit dynamics since it is the same with the Eqs. 1-5 in Sale 2024 and to emphasize the novel results in current study. We  included more explanations in the Appendix in the revised manuscript.
>
> > 2. What is the computational purpose of Hamiltonian sampling for a 1d density? The use of Hamiltonian dynamics (Aitchison), or non-normal dynamics (Hennequin) or other forms of annealed sampling (Savin) have historically been invoked in the service of sampling fast from complex distributions which are either heavily correlated or multimodal. I fail to see the logic of doing the same here. Please explain.
>
> Thanks for raising this important question. Our study tries to maximize the type of algorithms that can be implemented in a unified circuit dynamics and thus increase our understanding of the computational capacity of the circuit. We believe this is an important step when generalize the circuit to more complex computations, although the Hamiltonian appears to be over-kill for our current computational task of Gaussian cases, seems unnecessary, and its oscillations increase the MSE.
>
> A task that makes Hamiltonian essential could be the sequence __learning__ (briefly mentioned in line 482 in original manuscript), where a conventional algorithm is forward-backward (FB). How FB is implemented in the circuit remains unknown. Here we suggest the Hamiltonian oscillations involving forward and reverse sweeps may be a natural biological substrate for implementing FB. A direct implication of this is the circuit may use the Hamiltonian oscillation _period_ (probably regarded to theta oscillation, Fig. 3E-F) as a natural way to segment a continuous input stream into small segment. A similar idea was also proposed by Gupta, Nat. Neurosci., 2012 but we are unaware of any computational studies actually implementing it.
>
> > 3. Missing literature: several Kalman filtering implementations have been proposed in the lit. that do not seem referred to, e.g. Deneve et al 2007 "Optimal Sensorimotor Integration in Recurrent Cortical Networks: A Neural Implementation of Kalman Filters" which do not require the very coarse computational approximations due 1 sample per step updates.
>
> The paper was citated multiple times in our original manuscript (lines 40, 183 and 466).
>
> > 4. The math in the sampling section makes some implicit assumptions w.r.t. the map between the inferred latent variable z_t and the population responses of the neurons in the circuit u or r_e. Please clarify this map explicitly in the main text. More generally, the exact mapping between the neural dynamics elements and the dynamic inference implementation could be generally more explicit and better explained.
>
> We apologize that our dense manuscript may confuse the reviewers. We didn’t assume the mapping between the latent variable $z_E$ (Eq. 3a) and the population response. Instead, their relation was derived and discovered by rigorous math analysis of perturbative analysis and dimensionality reduction. It was briefly mentioned in Sec. 2.1, and all detailed math calculations can be found in Appendix Sec. D. We also expanded the explanations below Eqs. (3a-3b) to emphasize the relation between $z_E$ and the population response, and the mapping between circuit’s bump position dynamics and the theoretically defined sampling dynamics (line 241, 294) in the revised manuscript.
>
> > 5. Can you comment on the biophysical nature of the mechanism that controls the gain of the SOM subpopulation and how is that supposed to be calibrated to the true speed of the moving sensory stimulus?
>
> While the exact mechanism is beyond the scope of the paper, as mentioned in the discussion,  we hypothesize the speed-dependent SOM firing rate can be modulated via the VIP that is to be tested in future experimental studies. A recent study does show the SOM firing rate increase with speed, while it doesn’t show whether it is from the VIP modulation ( Kipper et al, bioRxiv 2025).
>
> For the speed calibration, we assume our circuit filters the latent stimulus that is generated by self-motion and then the motor cortex sends corollary discharge to notify sensory cortex (briefly mentioned in lines 361 and 452 in original manuscript). Although knowing the speed seems restrictive in our study, inferring the speeds corresponds to a learning problem since it is shared among latent stimulus over time. And most of previous studies assume the speed in latent transition is given.

---

> ### Author Response · Authors · 2025-11-21
>
> > 6. The faster sampling with faster speed is misleading as a mathematical statement" sampling speed refers to the autocorrelation of MCMC samples drawn from the same density whereas here we are talking about changes across samples driven by temporal changes in the mean of the density itself. this seems formally incorrect as a statement.
>
> Yes, we refer to the same mechanism you mention: the faster sampling means quicker decaying in the autocorrelation of MCMC samples. As explained in lines 387 – 391, the $\tau_E = \tau U_E$ controls the sampling time constant (the Table below Fig. 2). A larger speed requires larger SOM inhibition $U_{ES}$ (negative), then it reduces the $U_E$ (Eq. 4a). Surprisingly, this result is not purposely designed by us, but is the one we discovered from the circuit dynamics.
>
> > 7. Hamiltonian dynamics: can you please comment on the exact map between the variables of the sampler and the neural activity, in particular how is the momentum variable encoded in neural activity?
>
> Yes, your understanding is correct that we establish an analytical mapping from the circuit subspace dynamics (Eqs. 3a-3b) into the standard form of Hamiltonian sampling (Eq. 18), rather than manually designing a momentum dynamics and inserting it into the circuit. The tedious math derivations are presented in Appendix Sec. E2. Briefly, the momentum variable $p$ is defined as the linear mixture of $\delta z_s$, $\delta z_E$, and $\delta x_t$ in Eq. 17a, suggesting it is not regarded as a single circuit variable but a mixture of three circuit variables. And then our approach to verify the definition of momentum is through the math mapping into standard Hamiltonian form (Eq. 18). Note that the Hamiltonian sampling (Eq. 18) is not simply presenting this form, but we constrain the coefficients in Eq. 18 to satisfy the requirement of the same circuit without changing weights can realize Hamiltonian sampling under different uncertainties. Technically, this is done via comparing Eq. 18 with Appendix Eq. E24 (the standard form defined in theory).
> For the volume preservation, we think aligning Eq. 18 with Eq. E24 automatically proves this, although implicitly. Precisely speaking, Eq. 18 doesn’t preserve volume, since it is a mixture of Langevin sampling and Hamiltonian sampling (a form similar to Aitchison Plos Comp Biol 2016, Eqs. 16-17), where the Langevin sampling component will shrink the volume.
> For sampling the right distribution by Hamiltonian, we didn’t include the analytical error analysis as in Langevin case shown in Appendix Sec. E4 considering the length of the paper. If the reviewer think it is necessary, we are happy to include it in the revised manuscript, while its analysis will be more complex.
>
> > 8. What is the size of the networks numerically simulated in Fig2? is this the prediction of the asymptotic limit or the dynamics of a finite simulated network? Same question for Fig3
>
> The continuum limit is only considered in theory to facilitate math analysis, which is a convention for continuous attractor networks. In contrast, in simulation we use a finite number of neurons where our circuit has 180 excitatory neurons, and 180 SOM neurons for simulating Figure 2 and 3. Different network sizes scale the neuronal density $\rho$ (Eq. 4), then we only need to scale the neuronal weight $w_{XY}$ inversely the network can sample the same distribution.

---

> ### Author Response · Authors · 2025-11-21
>
> > 9. What makes the oscillations of hamiltonian dynamics (well documented in past work including Aitchison) have anything to do specifically with hippocampal activity? how is a simple kalman filter in 1d related to hippocampal computations?
>
> Thank you for this thoughtful question. The Hamiltonian in Aitchison’s paper is defined on the neural space that is like the $\mathbf{u}$ variable in Eq. 1a in our paper, while our sampling is defined on the stimulus $z_E$ subspace (Eq. 2 and 3a). Therefore, when regarding our circuit to be hippocampal CA3, our samples $z_E$ will be regarded as the internal or decoded spatial locations from place cells. Moreover, the 1d sample case corresponds that the animal is running on a linear track where the spatial location collapses into a 1d variable. Therefore, we imply a direct, comparison between our oscillatory sample and the decode spatial sequences in theta oscillations.
>
> Compared with the internal models included in Ujfalussy eLife 2016, our circuit model’s contribution is providing a link between mechanistic circuit model and the probabilistic internal model, which is largely missing in the field. Mathematically, our analysis suggests the internal representation in our circuit is most like the sampling scheme defined in Eq. 17 in Ujfalussy 2016 (briefly as Ujf16). Specifically, the theta phase $\phi$, sample $\tilde{x}$ in Ujf16 corresponds to the momentum $p$ phase and stimulus feature sample $\tilde{z}_t$ in our circuit model respectively. The encoding basis function $\phi_i$ in Ujf16 corresponds to the mean spatial population profile in our Eq. 2, and the Poisson process in Ujf16 is like our internal Poisson variability (Eq. 1a, last term).
>
> > 10. Discussion: based on the textbook definition of the term HMMs assume discrete latent variables whereas the speed of stimulus is continuous, why talk about alpha-beta type of inference and smoothing in HMMs specifically when 1. filtering is all you do for kalman inference and 2. a hierarchical kalman seems much more suited to the continuous nature of the latent variable in question. Please explain or correct.
>
> Thank you for giving us the opportunity to clarify this point. We consider a general definition of HMM with a continuous latent variable and perform filtering rather than the smoothing or forward–backward (FB) recursion. We are open to changing the name of HMM to the one suggested by the reviewer. We reference these algorithms in the discussion because learning unknown parameters such as latent speed is a natural extension, which filtering alone cannot provide.
>
> In our framework, the oscillatory structure of Hamiltonian sequential sampling offers a speculative but biologically grounded parallel: theta cycles contain alternating prospective and retrospective phases that sweep forward and backward in their latent space. This pattern is analogous to the temporal sweepings required by FB algorithms and suggests a potential circuit mechanism by which hippocampal theta sequences could support learning in future work. While the current model only does filtering, the dynamics point toward how theta oscillations might implement FB for parameter learning when speed is unknown. We have revised the discussion section to make this more evident (lines 524-530) .

---

### Official Review · Reviewer_tmYd · 2025-10-31

**Soundness:** 4
**Presentation:** 3
**Contribution:** 2
**Rating:** 4
**Confidence:** 3

**Summary:**

The authors introduce a canonical circuit that unifies Langevin and Hamiltonian sampling to infer either static or dynamic latent states
with various moving speeds.

It is shown that switching sampling algorithms and adjusting internal latent moving speed can be realized by modulating the gain of SOM neurons without changing synaptic weights.

When the circuit employs Hamiltonian sampling, trajectories resemble the decoded spatial trajectories from hippocampal theta sequences.

While the approach is elegant, it is hard to determine what the novel contributions are beyond previous work and to what extent the approach scales to more challenging real-world inference problems.

**Strengths:**

The paper provides an elegant theoretical framework with which Bayesian inference is related to processing in canonical biological circuits. The results show effective Langevin and Hamiltonian sampling in static and dynamic settings.

**Weaknesses:**

I find it hard to assess the novelty of this paper compared to the following earlier related work that is cited in the paper:

Eryn Sale and Wenhao Zhang. The bayesian sampling in a canonical recurrent circuit with a diversity of inhibitory interneurons. In The Thirty-eighth Annual Conference on Neural Information Processing Systems, 2024.

The current presentation is quite dense and it remains unclear if the current paper is an incremental improvement over this previous work. It would help if the authors make more clear in the text which contributions go beyond previous work. It would also help if previous work is used as a quantitative baseline to compare with to show where the current model provides significant improvements.

The approach is tested in simple toy settings. How does this generalize to more complex real-world Bayesian inference problems which may be high-dimensional and multimodal? It would help if the authors provide more extensive analyses to demonstrate the importance of their work for machine learning. Scaling results will also provide more certainty that the developed theory is used in biological systems.

Figure 2: it remains unclear what the quality of the static and dynamic inference is. Which metrics are used? Can baselines be introduced that demonstrate improvement over established approaches? Fig 2C suggests that the tracking of the true z is completely off. I may interpret this wrongly but more explanation/interpretation would be useful. The same for Figure 3. Panels D and F suggest that the sampling is far off from what would be ideal behaviour.

The authors make a connection between sampling and decoded spatial trajectories during hippocampal theta sequences. It remains unclear if it is valid to make a direct comparison between these two processes.

**Questions:**

The authors refer to canonical *cortical* circuits in their title and text. However, they make a link to biology by referring to hippocampal theta wave. However, hippocampus is a *subcortical* structure. This requires some thought.

In general text should be checked for typos and grammatical errors. There are quite a few sentences that are unintelligible. For example:

Line 101: fix Fig. A1B

Line 134: sentence needs some initial words.

Line 147: It has established theoretical approach  => fix

Line 208: Although the RBF with Gaussian case exists exact inference via Kalman filter

Line 264: remove ’t’ in this sentence

Line 448: Eq,

Line 460: Missing period

\xi_t in eqn 12 non-bold (consistency wrt other eqns)

---

> ### Author Response · Authors · 2025-11-21
>
> Thanks very much for the reviewer’s comments about the elegance of our theoretical framework and approach! Below we reply to your weaknesses and questions one by one.
>
> ## Weaknesses
>
> ### 1. Novelty
> Thank you for your comments. Since this was brought up by multiple reviewers, we have distinguished our paper with Sale 2024 in Global Official Comments “Novelty”.
>
> ### 2. Generalization into multivariate and multimodal cases.
> Please refer to the last section in our Global Official comments.
>
> ### 3. Figures
>
> For the parameter scans, the metrics are the mean square error (MSE). We include errors to show the transient tracking when feedforward input is off in Figure 2I.  We have revised Fig 2I where the shaded region is one standard deviation of the samples over 25 trails. The off-target effects are deviations where the feedforward input intensity is low as it is proportional to uncertainty; however, the true stimulus is still within the std. The circuit then rapidly re-enters and maintains tracking when feedforward input is higher.
>
> For Figure 3, the tracking is centered around the true stimulus but is not using SOM gain and speed where the MSE is minimized.  This is indicative of the stronger proportion of Hamiltonian sampling needed to induce oscillations similar to the forward and backward sweeps in the hippocampus also centered around the true stimulus.  We believe this is because the goals of networks in the brain are not only to track moving stimulus but make predictions about future paths and retrospectively review paths already taken, a well-studied phenomenon in the hippocampus (Wang, Science 2020).  We have clarified this point in Fig 3D.
>
> ### 4. Theta sequences: direct comparison to neual data
> Thank you for bringing this up.  While it is beyond the scope of the paper, we are developing quantitative comparisons between the hippocampal linear track data and our model such as speed-dependent sweep amplitude, step size, and spatial correlations, and compare these with model predictions.  We aim for exact analytical mapping between nonlinear circuits and algorithms for interpretability for machine learning models.
>
> ##  Questions
>
> > 1. The authors refer to canonical cortical circuits in their title and text. However, they make a link to biology by referring to hippocampal theta wave. However, hippocampus is a subcortical structure. This requires some thought. (Related to weakness)
>
> Thank you for bringing up this question.  The hippocampus is indeed a subcortical region, while the hippocampal CA3 region shares the similar canonical microcircuit structure as shown in Fig. 1A.
>
> > 2. General typos
>
> We thank the reviewer for their feedback.  We have checked and revised these issues in the new version.

---

> > ### Comment · Reviewer_tmYd · 2025-11-28
> >
> > Thanks to the authors for their detailed feedback. Upon reading the reviewer responses and author feedback I will retain my original assessment.

---

### Official Review · Reviewer_ougJ · 2025-11-02

**Soundness:** 3
**Presentation:** 2
**Contribution:** 3
**Rating:** 6
**Confidence:** 5

**Summary:**

This paper introduces neural circuit implementations of dynamical Bayesian inference (recursive Bayesian filtering) based on a sampling-based code. It explores Langevin and Hamiltonian non-equilibrium dynamics in E/I attractor networks with two types of inhibitory neurons, putatively corresponding to PV and SOM neurons. It suggests interesting and computationally novel roles for SOM neurons and theta oscillations.

**Strengths:**

The paper brings together sophisticated analyses of E/I attractor network dynamics with sampling theory, including advanced forms of sampling (Hamiltonian and Riemannian manifold), and makes novel links to biology.

**Weaknesses:**

- The main (and undiscussed) limitation of the model that it is restricted to a generative model in which the observations are the firings of a Gaussian-Poisson neural population encoding the relevant latent variable directly (and so the likelihood is a linear PPC). This prevents inference under any generative model of practical interest in which observations may be related to latent variables in more interesting way (e.g. inferring head direction from noisy self motion inputs and visual landmarks).

- The proposed algorithm essentially performs particle filtering with a single particle. The authors advocate for this essentially on biological plausibility grounds — but give the obvious concerns about computational accuracy short shrift by suggesting that a sufficient separation of time scales between latent and neural network dynamics will solve this problem automatically. I don't see how that is possible — there is potentially catastrophic information loss the moment the posterior is collapsed onto a single particle, and this information cannot be recovered no matter how soon the particle is updated. More broadly, I saw no calibration of the performance of the network (e.g. comparisons to standard recursive Bayesian filtering algorithms — particle-based or otherwise). Figures 2C & 3D do not allay my concerns about computational performance: the neural trajectories are far from the latent, and there is no attempt to extract error bars from neural activities to see if at least the latent is within error bars (or vice versa, if the neural trajectories are within the error bars of the exact posterior — see above comment about lack of calibration wrt e.g. exact inference). Figure 2I looks better, but it's dominated by prior knowledge baked into the network about latent variable drift, so it's unclear how online inference by the circuit actually contributes here.

- The links to biology are interesting but somewhat superficial. In particular, the first-order effect of theta oscillations on hippocampal activity is a strong modulation of overall firing rates of both E and I cells (with different characteristic profiles and preferred phases of firing for E, PV, and SOM cells [the latter aka O-LM cells in the hippocampus]). Also highly relevant for the current model is that theta frequency shows robust modulation by movement speed. Are any of these effects borne out in the model?

- The presentation is somewhat confusing. For example, based on section 3.2, in the static case, p(z_{t+1} | \tilde{z}_t) is a delta on the previous location of the particle (from Eq.5), which can thus only stay at the same place in the next time step (based on Eq.8). Then, in section 3.3, for the Langevin sampling variant (also shown in Fig.2A), it is stated that tau_L \propto \Omega_t is used (from Eq.13). However, from Eq.9 \Omega_t→\infty in the static case (because \Lambda_z→\infty) which would again suggest infinitely slowed down dynamics. Despite all this, in Fig.2A this is clearly not the case.

- There is ambiguity as to whether the fact that the proposed "canonical circuit unifies Langevin and Hamiltonian sampling" is itself novel or not. The abstract makes it sounds it is novel, but later it seems that the static case has already been covered by Sale & Zhang, 2024.

- There are a number of grammatically incorrect sentences. E.g. "Although the RBF with Gaussian case exists exact inference via Kalman filter"

**Questions:**

- Is there a meaningful way to quantify the performance of the circuit models and compare it to relevant baselines (as well as to one another)?

- Can the model "predict" theta modulation, and different theta phase preferences, of different E and I cell types, or the modulation of theta frequency by running speed?

- Could you discuss more explicitly novelty wrt Sale & Zhang, 2024?

---

> ### Author Response · Authors · 2025-11-21
>
> We very much appreciate the reviewer’s appreciation of our rigorous theoretical analysis of nonlinear circuit dynamics to investigate its sequential sampling algorithms for dynamic inference. Below we reply to your weaknesses and questions one by one.
>
> ## Weaknesses
>
> ### 1. Generative Model
> Thank you for the discussion. This is a topic several reviewers wish to visit; therefore, we have added it to the Global Official Comments. Please refer to the last two sections there.
>
> ### 2. Sampling Framework
>
> Thank you for allowing us to clarify.  In Eq. 7-8, the predictive posterior distribution (the integration term) is actually a __hybrid__ representation, rather than a pure sampling-based representation from a single sample or particle that collapses into a delta function as in particle filtering. Specifically, the predictive posterior in Eq. 7 is approximated by the distribution $p(z_{t+1}  |\tilde{z}\_t)$ in Eq. 8, which is a parametric Gaussian distribution that maps to the parametric recurrent E interactions $u_{EE} = W_{EE} *r_E$, but its mean parameter depends on a single sample $\tilde{z}\_t$ represented by $r_E$. Then, recurrent connections $W_{EE}$ effectively transforms the $\tilde{z}\_t$ into $p(z_{t+1}|\tilde{z}\_t)$ with its coding mechanism similar to the one presented in Zhang, Nat. Comms 2023.  Therefore, there will not be catastrophic information loss. We expanded the text after Eq. 8 to avoid this confusion in the revised manuscript (lines 227-233).
>
> ### Figures with your concern of computational performance
> Regarding the calibration question, Figures 2C and 3D were intended to highlight specific circuit mechanisms such as how recurrent excitation $W_{EE}$ affects the mean square error (MSE) in Fig. 2C,  and how the oscillatory samples produced by Hamiltonian sampling. Fig. 2I includes the 2 standard deviation  of the instantaneous posterior (rather than the std of the samples over trials)   as the shaded region. In the revised manuscript, we have repeated this simulation where the shaded region is 1 std of the samples over 25 trials to clarify this point.  Even with the off-target deviations where feedforward input intensity is small, the true stimulus is within the shaded region (Fig A8).  The seemingly off-target deviations temporary reflects the transient dynamics due to changing feedforward input intensity. Even when the observed trajectory appears to lose tracking of the true stimulus and lay beyond the shaded regime, the circuit rapidly re-enters and maintains tracking accuracy when referring to the right most part of the trajectory.
>
> For Figure 3, the tracking is centered around the true stimulus, but is not using SOM gain on the black line in Fig. 3A with minimized MSE. This serves to illustrate the oscillations associated with stronger proportion of Hamiltonian sampling and to compare it with forward and backward sweeps in the hippocampus theta sequences.  We believe this is because the goals of networks in the brain are not only to track moving stimulus but make predictions about future paths and retrospectively review paths already taken, a well-studied phenomenon in the hippocampus (Wang, Science 2020). We also believe the oscillations serve a role in learning which can be conducted through the forward-backward algorithm (see our reply to weakness #1 to reviewer mCaV).  This clarification has been added to the Fig 3D   figure caption..
>
> ### 3. Link to biology
>
> The theta sequences in Fig. 3E refer to the decoded spatial trajectories from place cells during theta oscillations, and our Hamiltonian sequential samples (Fig. 3F) resemble a similar structure. Due to the space limit, we don’t focus on the firing rate oscillations of single neurons. Due to the math similarity between our model and Chu, eLife 2024, we believe our circuit model can reproduce the theta firing rate.
> The speed-dependent theta is well documented in experimental studies (Whishaw Behav. Biol, 1973, Gupa Nat Neuro 2012; Hinman J. Neurophysiol, 2011; Hinman Neuron 2016; Jeewajee Hippocampus 2008; McFarland J. Comp. Physiol, 1975;Winter Curr Biol, 2015) . It is straightforward to see larger inhibition will increase the oscillation strength from circuit dynamics perspective. The novel implication from our model is the speed-dependent theta can be a computational consequence from Hamiltonian sequential sampling, which, to our best knowledge, has not been proposed before. We revised the paragraph of Experimental prediction in the Discussion section in the revised manuscript (lines 486-493).

---

> ### Author Response · Authors · 2025-11-21
>
> ### 4. Transition probability
>
> We apologize for the confusion and thank the reviewer for pointing this out. We have revised our manuscript by clearly defining the target static posterior in the revised Sec. 4.2
> You are correct. When $\Lambda_z \rightarrow \infty$, the $ p(z_{t+1} | \tilde{z}_t)$ becomes a delta function, then the precision of instantaneous posterior $\tau_t(z_t) \equiv p(z_t|r_F(1:t)) $ will be $\Omega_t  = t * \Lambda_F$ that linearly increases with time. Hence, the instantaneous posterior is effectively doing evidence accumulation.
> In contrast, the static posterior is defined as $p(z|r_F)$, corresponding to the circuit doesn’t accumulate past inputs, and just generates random samples to approximate the time-invariant posterior $p(z|r_F)$.
>
> ### 5. Novelty
> Please refer to the section “Novelty” in the global official comment.
>
> ### 6. Grammar
>
> Thank you for the corrections, we have performed a thorough proofread and corrected the typos in the revised version.
>
>
> ## Questions
>
> > 1. Is there a meaningful way to quantify the performance of the circuit models and compare it to relevant baselines (as well as to one another)?
>
> The present study uses mean square error (MSE) to quantify the performance of the sequential sampling in the circuit, however, we haven’t systematically compared our performance with other models, e.g., the deterministic circuit model performing Kalman filter in Deneve 2007 and Wilson 2009, due to space limit.
> Notably, our suggested comparison is on how the circuit dynamics implement various algorithms and/or the requirements of different algorithms on circuit operations and components. For example, our result suggests that the sampling can simplify the complexity of circuit implementation. Please refer to the section “Sampling simplifies the complexity of circuit implementation” in the global official comments. Once the field builds clear mappings between circuit dynamics and algorithms, the performance quantification can directly use the results widely tested in ML society.
>
> > 2. Can the model "predict" theta modulation, and different theta phase preferences, of different E and I cell types, or the modulation of theta frequency by running speed?
>
> Thank you for your question.  Yes, our network with E and PV/SOM interneurons predicts speed-dependent theta oscillations as observed in previous experiments with 7arger sweep amplitude and oscillation frequency studies (Whishaw Behav. Biol, 1973, Gupa Nat Neuro 2012; Hinman J. Neurophysiol, 2011; Hinman Neuron 2016; Jeewajee Hippocampus 2008; McFarland J. Comp. Physiol, 1975; Winter Curr Biol, 2015) with a hypothesized mechanism of the increased SOM gain. Meanwhile, the SOM gain is observed increases with speed in recent studies (Kipper bioRxiv 2025).
> For the theta phase preferences, our model naturally predicts the theta phase in SOM neurons will be delayed than the one in E neurons.
>
> > 3. Could you discuss more explicitly novelty wrt Sale & Zhang, 2024?
>
> Since this was brought up by multiple reviewers, we have added the discussion in a global reply.

---

### Official Review · Reviewer_mCaV · 2025-11-05

**Soundness:** 3
**Presentation:** 3
**Contribution:** 2
**Rating:** 4
**Confidence:** 3

**Summary:**

This paper extends the work of Sale & Zhang (2024) on Bayesian sampling in canonical cortical circuits to the problem of dynamic inference. The authors propose that the same circuit, consisting of excitatory neurons, and parvalbumin/somatostatin interneurons, can implement both Langevin and Hamiltonian sampling to infer either static or dynamic latents. The key mechanism enabling this flexibility is the modulation of SOM neuron gain ($g_{S}$​), which the authors decompose into two components: a speed-dependent gain that encodes the stimulus velocity, and a “switching” gain that controls the ratio between Langevin and Hamiltonian sampling. The authors demonstrate through theoretical analysis of the nonlinear circuit dynamics that the circuit can track moving stimuli without modifying synaptic weights, and show that Hamiltonian sampling trajectories produce oscillations resembling hippocampal theta sequences.

**Strengths:**

Originality: the paper makes a theoretical contribution from static to dynamic inference within the Sale & Zhang (2024) sampling framework. While the foundational circuit architecture is borrowed from Sale & Zhang (2024), the application of this model to dynamic inference represents conceptual progress. They provide a novel functional role for the gain of the SOM neurons in the context of dynamic stimuli, and offers an original perspective on the functional role of oscillations in hippocampal theta sequences.

Quality: the mathematical analysis is rigorous and follows established methods for analyzing continuous attractor networks. The authors provide detailed perturbative analysis, eigenmode decomposition, and explicit mappings between circuit dynamics and Langevin and Hamiltonian sampling algorithms. The derivations connect circuit parameters to posterior distributions in a principled way.

Clarity: the paper is generally well-structured. The progression from static to dynamic inference is well explained, and the figures are effective in illustrating the main concepts and results. The supplementary materials provide thorough derivations.

Significance: the potential unification of different sampling algorithms within a single circuit architecture is conceptually appealing. The model makes concrete predictions about the relationship between SOM gain, stimulus speed, and neural oscillations, and if these are validated experimentally, this work could offer insights into how cortical circuits implement probabilistic inference in dynamic environments.

**Weaknesses:**

1. The most fundamental weakness in the paper is in what constitutes “dynamic inference”. The circuit already receives the velocity $v$ as input via SOM gain modulation (Eq. 16, Fig. 2G). This undermines the claim of performing dynamic inference, since the primary goal of dynamic inference is typically to *infer* the latent dynamics from noisy observations. If the velocity is provided a priori, the circuit is merely tracking a stimulus with known dynamics rather than performing inference over hidden states. Similarly, the transition precision $\Lambda_z$​ is hardcoded into the recurrent weights $w_{EE}$​ (Eq. 14b). The authors argue this is necessary to implement adaptive step sizes, but it means the circuit cannot adapt to varying environmental statistics without synaptic plasticity. This is at odds with the authors’ argument that gain modulation is preferred over synaptic weight changes because it operates at faster timescales. This rigidity limits the generality of the approach compared to more flexible inference algorithms.
2. Moreover, restrictive assumptions limit the generality of this approach. The framework relies heavily on three assumptions. First, the model assumes that feedforward inputs have Gaussian tuning curves (Eq. 1e) leading to Gaussian likelihoods (Eq. 6). However, real sensory likelihoods are often non-Gaussian and multimodal, and one of the purported strengths of sampling-based approaches is that they can represent arbitrary distributions. The authors do not address how the circuit would handle non-Gaussian inference problems. Second, the model assumes a 1D ring attractor, and its specific eigenmode structure is essential for the perturbation analysis (Eqs. D15-D16). It's unclear how the approach would scale to higher-dimensional feature spaces without this structure. While the authors show a 2D extension (Fig. A4) that couples two ring attractors, this is still a rather restrictive latent structure which presumably not all canonical circuits possess. Finally, the analysis uses uniform priors throughout, which means the posterior is essentially a scaled likelihood. This eliminates one of the key computational challenges of Bayesian inference, which is to show that the prior can reflect the statistics of its inputs. The authors do not demonstrate that the circuit can implement informative non-uniform priors or that it can flexibly switch between different prior distributions.
3. I am not sure about the necessity of sampling in your circuit model if not to encode posterior uncertainty. The noise in the circuit dynamics is framed as enabling exploration, but the posterior uncertainty is entirely determined by the feedforward input rate $R_F$​ (which controls likelihood precision $\Lambda_F$​). The neural variability does not represent posterior uncertainty in the way that sampling-based models typically propose. The authors state that "a single snapshot of $\mathbf{r}_F(t)$ ​parametrically conveys the whole stimulus likelihood" (Section 3.1), meaning a population vector readout is sufficient. In this case, it is unclear what computational advantage sampling provides over simpler population coding schemes like probabilistic population codes (PPC), which can also perform Bayesian inference with linear readouts but without the complexity of maintaining sampling dynamics.
4. I believe there is insufficient comparisons with alternative approaches. The authors briefly mention that deterministic inference circuits require "complicated nonlinear functions to implement marginalization," which their sampling approach avoids. However, this advantage is specific to the problem structure (Gaussian distributions, linear-Gaussian dynamics). Moreover, the authors do not compare computational costs, convergence speed, or accuracy with deterministic approaches like Kalman filters or variational inference. Overall, the claim that their sampling framework provides advantages is not substantiated with quantitative comparisons.
5. There is limited biological justification for key mechanisms in the model. While the authors mention that VIP neurons might modulate SOM gain to convey self-motion signals, this remains speculative. The assumption that the motor system provides precise, instantaneous velocity information to sensory cortex requires substantial justification. Moreover, the paper assumes that SOM neurons do not receive feedforward input (necessary for Hamiltonian sampling), but this constraint may not hold across all cortical areas and contradicts some anatomical evidence.

**Questions:**

- How would the circuit handle time-varying or unknown velocities? How does the circuit adapt when transition statistics change?
- How would the circuit handle non-Gaussian likelihoods, which are common in real sensory processing? Can you provide numerical experiments or extensions of your framework to cases where the Gaussian assumption breaks down?
- Given that the likelihood can be read out with a population vector (linear decoder), what specific computational advantages does sampling provide over probabilistic population codes (PPC) in your framework? Can you provide quantitative comparisons (e.g. in terms of inference accuracy or speed)?
- If the transition probabilities are Gaussian and the likelihood is a Gaussian function, as in your setup, then the recursive Bayesian filtering algorithm reduces to the Kalman filter, where the instantaneous posterior is also Gaussian. In this case, what is the point in approximating this quantity with sampling when an exact solution is attainable?
- How sensitive is the circuit to mismatches between the assumed velocity $v$ (encoded in SOM gain) and the true stimulus velocity? What happens when there are errors/noise in gain modulation?
- How would the circuit acquire the precise weight configurations required for sampling (Eqs. 14, 19)? Do you think these synaptic weights are learned?

---

> ### Author Response · Authors · 2025-11-21
>
> We thank the reviewer for positive comments about our algorithmic understanding of sequential sampling in circuit via theoretical approach! Below we reply to your weaknesses and questions.
>
> ## Weaknesses
>
> ### 1. Dynamic Inference
> Thank you for raising this key point. The speed $v$ is a parameter shared among latent states over time, and hence inferring the speed corresponds to a __learning__ problem in machine learning, rather than inference. Even when the speed is given to the network, the circuit is estimating latent hidden state sequences.  Meanwhile, we’d like to emphasis that previous dynamic inference neural circuit studies also focused on the case that the transition parameters, including speed, are either fixed or given, including Kutschireiter PNAS 2003, Eq. S1, Section SI 1A.
> How neural circuits learn parameters in an HMM is beyond the scope of the present study and is not well addressed in the field. Our Discussion mentions our conjecture that Hamiltonian sequential sampling can be beneficial for sequential learning using methods such as the forward-backward (FB) algorithm. We suggest oscillations in Hamiltonian sampling may naturally implement FB algorithms, and the oscillation cycle may automatically segment a long sequence into many small sequence segments for learning. Considering the similarity of Hamiltonian sequential sampling and hippocampal theta sequences, and the importance of theta in memory formation and retrieval, our framework has the potential to provide computational insights into the role of theta sequences during learning. We expanded this discussion in the revised manuscript since it allows an extra page (lines 525-530).
>
> ### Representing transition precision in PV gain
> We appreciate the acknowledgement of utilizing fast gain modulation to switch the hyperparameters of algorithms employed by circuit dynamics. The transition precision is directly determined by recurrent E input strength, $U_{EE}\propto w_{EE} R_E$ (lines 314-317 ), which depends on the recurrent weight $w_{EE}$ and the global inhibition strength in divisive normalization (Eq. 1b). Therefore, another possibility of modulating transition precision is modulating the gain of PV neurons by introducing a gain factor, $g_P$, into PV (Eq. 1b; PV). Then a larger $g_P$ leads to smaller $U_{EE}$ with less transition precision. In the newly added supplementary figure, we also show this is plausible (Fig. A8). Then the fast PV gain modulation also provides a way to change the transition precision in fast time scale. We also added a paragraph in the Discussion in the revised manuscript (lines 479-485) to discuss this possibility.
>
> ### 2. Assumptions
> Thank you for giving us the opportunity to clarify. Since this feedback is common among reviewers, we address this into the global Official Comments, where we explain the novelty and significance of our study, and how we extend our framework into more complicated case.
> Here we’d like to clarify a few things for the reviewer.
>
> First, our coupled circuit can easily generalize to higher dimensional static or dynamic posterior sampling, where the number of circuit motifs in the coupled circuits is determined by the posterior dimension. This approach is plausible as illustrated in earlier studies, e.g., Zhang, J. Neurosci., 2016 (Fig. 7).
>
> Second, for the multivariate static posterior sampling in coupled circuits, the recent work suggests the circuit stores a non-uniform associative prior (Sale, NeurIPS 2024; Eq. S85).
>
> Third, for dynamic inference in the present study, the predictive posterior from the last time step (Eq. 7, integration term in RHS) becomes an __effective prior__ in the next time step. This effective prior is non-uniform given the non-trivial transition probability in Eq. 5, and thus the dynamic posterior is not simply a scaled likelihood but needs to be computed via filtering (Eq. 7). And we have analytically derived how this effective prior or transition probability is represented in the circuits, including transition precision (Eq. 14b and Fig. 2D) and transition speed (Eq. 16 and Fig. 2G).
>
> ### 3. Sampling framework
> For dynamic inference, the uncertainty of instantaneous posterior (Eq. 7; Eq. 9 $\Omega$) is __jointly__ determined by the feedforward input and the transition probability (Eq. 5) stored in the circuits. Therefore, the population vector can only get the likelihood from feedforward input but cannot recover the posterior due to the marginalization in filtering (Eq. 7). The sequential sampling in the circuit draws samples (Eq. 8) and this variability directly reflects posterior uncertainty, consistent with sampling-based representation. Further discussion of the sampling framework is included in the global reply.

---

> ### Author Response · Authors · 2025-11-21
>
> ### 4. Alternative approaches
>
> We’d like to elaborate on how sampling can simplify circuit implementation, while the length constraint of submission limits our explanation. An important issue we notice from our study is simpler algorithms, e.g., Kalman filter appears simpler than sequential sampling, doesn’t necessarily mean simpler circuit implementations.
>
>  Even the simple Gaussian/ von Mises distributions in computations needs complex operation/approximations in circuit implementation, and we are concerned about how we utilize the circuit model to do more complicated computation / distributions. Therefore, we shift to sampling and aim to utilize the flexible representation of sampling to simplify circuit implementation. Indeed, our study shows sampling only requires linear stochastic dynamics in the subspace in neural circuits. Although we only consider a simple Gaussian form that makes the sampling unnecessary in computation, we believe building an exact, analytical mapping between circuit dynamics and sampling algorithm paves the way to utilize this sampling circuit to achieve more complicated tasks in the future (see our reply in global Official Comment), which is also important to obtain the interpretability of circuit computations.
>
> In reply to quantitative comparison, it is straightforward to compare the performance at the algorithmic level, e.g., comparing the Langevin sequential sampling (Eq. 10) with the Kalman filter (Eq. 1-3 in Wilson, NIPS 2009). However, it is technically challenging to quantitatively compare the computations between neural circuit models executing different algorithms, e.g., comparison between our circuit model with the one in Wilson 2009. This is because 1) different studies assume different generative models; and 2) The algorithms are usually embedded into the recurrent circuit dynamics, and it is very hard to change the circuit components to switch circuit algorithms or change hyperparameters to make a fair comparison. Certainly, we are open to further suggestions from the reviewer for a direct, fair comparison between computations across circuit models.
>
> ### 5. Biological Plausibility
>
> Thank you to the reviewer for asking about the discussion. A recent experimental study suggests the SOM firing rate does increase with the running speed (Kipper et al, bioRxiv 2025), supporting our claim. Nevertheless, it still lacks direct evidence that the increased SOM firing rate with speed is modulated via the VIP, and it is largely a hypothesis to be tested in the future.
> For our assumption of the circuit receiving the speed signal, please refer to our reply to your 1st question, as this setting is the same as earlier studies in the field.
> For the question regarding no direct feedforward input to SOM, this is indeed supported by neuroanatomical studies that SOM receive fewer feedforward synapses than other neuron types (e.g., Fig. 2B, L4 Pyr to L2/3 SSt or L5 SSt in Campagnola et al., 2022 ). Interestingly, Sale, NeurIPS 2024 computationally derived Hamiltonian sampling requires no feedforward input to SOM. This suggests the scarcity of  feedforward input to SOM might be not a coincidence, but a need for circuit computations.

---

> ### Author Response · Authors · 2025-11-21
>
> ## Questions
>
> Thank you for your questions, please see our responses above for questions 1-4.
>
> > 5. How sensitive is the circuit to mismatches between the assumed velocity  (encoded in SOM gain) and the true stimulus velocity? What happens when there are errors/noise in gain modulation?
>
> Thank you for your question.  We used the mean square error (MSE) to quantify the accuracy of circuit’s estimate.  In Fig 2G and 3A, we show there exists a linear manifold for the network to minimize MSE.  Deviating from that line manifold due to mismatched SOM gain with speed increases the MSE but the MSE landscape overall is flat, suggesting the circuit is not quite sensitive to the mismatch.
>
> > 6. How would the circuit acquire the precise weight configurations required for sampling (Eqs. 14, 19)? Do you think these synaptic weights are learned?
>
> Thank you for the opportunity to clarify.  For these parameters, these weights are learned when the animal encounters a novel environment through synaptic plasticity.  Our above reply also suggests the Hamiltonian sequential sampling may facilitate sequence learning.

---

### Author Response · Authors · 2025-11-21
**Global Rebuttal**

We thank reviewers for their thoughtful comments. We appreciate the recognition of the rigor and our contribution to understanding the algorithmic interpretation of nonlinear circuit dynamics. Below we address common concerns.

### Main contribution
Understanding algorithms embedded in nonlinear recurrent circuits remains challenging. Using a biologically plausible and analytically tractable circuit, we derive exact mappings from the circuit dynamics (Eq. 1a-1e) to sampling dynamics (Eq. 3a-3b, Sec. 3.2). This is rare in Bayesian neural circuit studies and allow us to open the black box of nonlinear dynamics to “see” the algorithm.

We show for the first time that circuit dynamics with fixed weights can implement __both__ Langevin and Hamiltonian sampling algorithms for static __and__ dynamic inference via fast SOM gain modulation.  This significantly advances our understanding of circuit computations. We further show that Hamiltonian sequential sampling produces dynamics resembling hippocampal theta sequences, suggesting a computational role for theta.

 ### Novelty

Although the circuit equations match Sale 2024 (Eq 1-5 Sale, Eq 1-4 current), the __algorithmic understanding__ obtained here is novel:

1) We study dynamic inference with a non-trivial transition probability (Eq. 5), whereas Sale 2024 studied static inference.  We show that the dynamic circuit can be naturally backwards compatible with static inference, enabling multiple tasks within one architecture.
2) We analytically identify sequential sampling algorithms (Sec. 3-2-3.3) and map them to circuit dynamics (Sec 4.1, 4.3).  Sequential sampling is absent in Sale 2024.  Although Eq. 10 resembles static sampling (Eq. 14 Sale 2024), the subscript $t$ corresponds to non-equilibrium vs. equilibrium sampling. We think this “similarity” is what enables flexible switching of the computational tasks. Importantly, this similarity (Eq. 10) relies on evaluating the score of $\pi_t(z)$ at the previous sample $\tilde{z}_{t-1}$, otherwise an extra term in Langevin sequential sampling appears and changes dynamics.
3) We reveal a new function of SOM gain. It sets the internal transition speed (rows in Fig. 3G) in addition to switching sampling algorithms as in Sale 2024 (columns in Fig. 3G).

### Sampling simplifies circuit implementation
While it appears unnecessary to develop a sampling algorithm for Gaussian posteriors with close-form solutions where deterministic computations, e.g., Kalman filter, may be faster, simpler, and more accurate, deterministic algorithms may not have simpler circuit implementation. A direct implication from our work is _the sampling can simplify the neural circuit implementation_, a benefit of sampling not well recognized in the field.

Deterministic filters (e.g., Kalman) require nonlinear operations (Eq. 11, Wilson, NIPS 2009) and are distribution dependent, e.g., Kutschireiter PNAS 2023 considered von Mises cases and derived another nonlinear form (Eq. 10). This complicates biological implementation and requires case-by-case circuit approximations. Then, research seeks to find required nonlinear operations in circuits, e.g., dendritic nonlinearities in Ujfalussy 2016.

In contrast, the nonlinear operation in deterministic circuit for marginalization is replaced by linear, stochastic neural dynamics (Eqs. 8,10). Although demonstrated for the Gaussian case, establishing an exact mapping between sampling algorithms and circuit dynamics enables us to handle more complex tasks in the future, especially sampling is flexible in approximate complex distributions.

### Complex generative models
Our study focuses on revealing the circuits' computational algorithms and tasks, by utilizing a feedforward input model parametrically representing Gaussian likelihood and a non-trivial latent transition probability. Extending to complex models is of high interest:

- Multivariate and multimodal. Prior work with the same circuit demonstrated bimodal distribution sampling (Fig S2, Sale 2024) and preliminary perturbative analysis for the bivariate case.

- Non-Gaussian. A von Mises likelihood requires switching the Gaussian recurrent kernel (Eq. 1d).  In general, we can use the network to sample multivariate distributions with each marginal belonging exponential family distributions where natural parameters will determine the recurrent kernel profile (Zhang Nat. Comms 2023 and Ma, Nat Neurosci, 2006). By contrast, as mentioned earlier, the representation of multivariate exponential family distribution in deterministic circuits require complex, nonlinear operations.

- General models. As in VAEs or diffusion models, encoders/decoders have been used to map complex data distributions into simple latent Gaussians. Our circuit may serve as the latent sampler. Similar connections were suggested by Peng, NeurIPs 2025, though without the Bayesian circuit interpretation. This may offer a biologically plausible route toward NeuroAI.

---

### Author Response · Authors · 2025-11-26

We would like to thank all the reviewers for their time and effort in reviewing our manuscript. With the discussion period ending next week, please let us know if we can address any further comments or concerns.

---

### Note · Authors · 2026-05-02

I have read and agree with the venue's withdrawal policy on behalf of myself and my co-authors.

---

### Meta-Review · Area_Chair_nkZh · 2026-01-06

**Summary:**

This paper focuses on the hypothesis that canonical cortical microcircuits perform Bayesian inference by sampling and extends the model of Sale & Zhang (2024) from a static to a dynamic setting. Their model proposes that such circuits could perform both Langevin and Hamiltonian sampling and switch between them without changing synaptic weights.

While the reviewers appreciated the theoretical rigor, mathematical analysis and novel hypothesis proposed for SOM gain functions, they were concerned about the restrictive assumptions (Gaussian model, closed-form solutions) and limited novelty compared to Sale & Zhang 2024, biological plausibility and practical significance.

The paper received three rejects (including one strong reject) and only one borderline accept.

**Reviewer Concerns:**

While the reviewers appreciated the theoretical rigor, mathematical analysis and novel hypothesis proposed for SOM gain functions, they were concerned about the restrictive assumptions (Gaussian model, closed-form solutions) and limited novelty compared to Sale & Zhang 2024, biological plausibility and practical significance.

The paper received three rejects (including one strong reject) and only one borderline accept.

**Reviewer Scores:**

I suspect the reviewers are unlikely to have changed their reviews significantly.

---

### Decision · Program_Chairs · 2026-01-26

Reject